# The human TRPA1 intrinsic cold and heat sensitivity involves separate channel structures beyond the N-ARD domain

Lavanya Moparthi [1,2] ✉, Viktor Sinica[3], Vamsi K. Moparthi[4], Mohamed Kreir [5], Thibaut Vignane [6], Milos R. Filipovic[6], Viktorie Vlachova [3] & Peter M. Zygmunt [7] ✉

TRP channels sense temperatures ranging from noxious cold to noxious heat. Whether specialized TRP thermosensor modules exist and how they control channel pore gating is unknown. We studied purified human TRPA1 (hTRPA1) truncated proteins to gain insight into the temperature gating of hTRPA1. In patch-clamp bilayer recordings, Δ1–688 hTRPA1, without the N-terminal ankyrin repeat domain (N-ARD), was more sensitive to cold and heat, whereas Δ1–854 hTRPA1, also lacking the S1–S4 voltage sensing-like domain (VSLD), gained sensitivity to cold but lost its heat sensitivity. In hTRPA1 intrinsic tryptophan fluorescence studies, cold and heat evoked rearrangement of VSLD and the C-terminus domain distal to the transmembrane pore domain S5–S6 (CTD). In whole-cell electrophysiology experiments, replacement of the CTD located cysteines 1021 and 1025 with alanine modulated hTRPA1 cold responses. It is proposed that hTRPA1 CTD harbors cold and heat sensitive domains allosterically coupled to the S5–S6 pore region and the VSLD, respectively.

Several transient receptor potential (TRP) channels are involved in thermosensation allowing organisms to sense a range of temperatures between noxious cold and noxious heat[1–5]. However, only a few of these thermoTRPs (TRPV1, TRPV3, TRPM8, and TRPA1) have been shown to possess pronounced inherent temperature sensitivity[6–10], of which the human TRPA1 (hTRPA1) responds to both cold and heat[11]. The bidirectional thermosensitivity of hTRPA1, and the findings that chimeras of rat TRPM8 and TRPV1 as well as single point mutations of the mouse TRPA1 caused opposite temperature responses[12,13] may support the idea that each thermoTRP is both cold and heat sensitive[14]. A dual heat and cold sensitivity could involve a single or several specific cold and heat sensing domains that undergo protein denaturation

or otherwise are allosterically coupled to the channel gate[14–17]. Although single particle cryo-electron microscopy (cryo-EM) studies of TRP channels have advanced the knowledge of their structure–function relationship[10,18–23], we are only at the beginning of understanding which of the TRP channels are true thermosensors, and whether specialized thermosensor modules within the proteins exist and how they are linked to channel pore gating.

We have previously shown that hTRPA1 and *Anopheles gambiae* TRPA1 (AgTRPA1) are intrinsically cold and heat activated, respectively, with and without the N-terminal ankyrin repeat domain (N-ARD) and part of the pre-S1 region[7,8]. Furthermore, we found that hTRPA1 is also heat activated and its cold and heat sensitivity may involve

---

[1]Wallenberg Centre for Molecular Medicine, Linköping University, SE-581 83 Linköping, Sweden. [2]Department of Biomedical and Clinical Sciences (BKV), Faculty of Health Sciences, Linköping University, SE-581 83 Linköping, Sweden. [3]Department of Cellular Neurophysiology, Institute of Physiology of the Czech Academy of Sciences, 142 20 Prague, Czech Republic. [4]Department of Physics, Chemistry, and Biology, Division of Chemistry, Linköping University, SE-58183 Linköping, Sweden. [5]Janssen Research & Development, Division of Janssen Pharmaceutica N.V., Turnhoutseweg 30, 2340 Beerse, Belgium. [6]Leibniz-Institut für Analytische Wissenschaften-ISAS-e.V., Bunsen-Kirchhoff-Straße 11, 44139 Dortmund, Germany. [7]Department of Clinical Sciences Malmö, Lund University, SE-214 28 Malmö, Sweden. ✉e-mail: lavanya.moparthi@liu.se; peter.zygmunt@med.lu.se

different protein conformations and possibly separate cold and heat sensors[11,24]. We reasoned that depending on the cellular environment including the channel redox state, thermoTRPs can adopt various conformations of which some are sensitive to cold and others to heat[11]. Although TRPV2 was not considered as a redox-sensitive TRP channel[25], we recently revealed mammalian TRPV2 as oxidation-sensitive, but with another redox profile than, e.g., TRPA1[26]. Thus, hidden thermoTRP redox properties may be of fundamental importance when exploring their intrinsic polymodal sensory properties and physiological/patophysiological role as well as druggability for treatment of e.g., pain, and inflammation-induced and immune-mediated diseases.

In this study, we have further investigated the intrinsic cold and heat properties of hTRPA1 beyond the N-ARD, and also with regard to the channel redox state. We found that the heat sensitivity is intact in purified Δ1–688 hTRPA1 (without the N-ARD) but lost in Δ1–854 hTRPA1 (without the N-ARD and S1–S4 transmembrane domain), which displayed an increased cold sensitivity. Different lipid bilayer-independent conformational changes of purified Δ1–688 hTRPA1 and Δ1–854 hTRPA1 caused by cold and heat were disclosed by measuring their intrinsic tryptophan fluorescence. Furthermore, cold and heat responses of purified Δ1–688 hTRPA1 and Δ1–854 hTRPA1 were lost in the presence of the thiol-reducing agent TCEP, and cold responses of

hTRPA1 subjected to single point mutations of the cysteines 856, 1021 and 1025 were affected when expressed in HEK293T cells. It is proposed that heat sensitivity is dependent on the voltage sensing-like S1–S4 transmembrane domain (VSLD) and the C-terminus domain distal to the transmembrane pore domain S5–S6 (CTD), whereas the cold sensitivity is dependent on the S5–S6 pore region and CTD.

## Results

### Purification and biochemical characterization of Δ1–854 hTRPA1

We have previously purified hTRPA1 with and without its N-terminal ARD (Δ1–688 hTRPA1, 95% purity) for functional and structure analyses[7,11]. Here, we have further deleted hTRPA1 of its S1–S4 transmembrane domain (Δ1–854 hTRPA1) (Fig. 1a). The purity of Δ1–854 hTRPA1 was estimated to be at least 80% without any substantial single impurity (Fig. 1b, c). Gel filtration (Fig. 1d) and circular dichroism (CD) spectroscopy (Fig. 1e) were used to determine the tetramerization and folding of the protein. As shown by the chromatogram, Δ1–854 hTRPA1 eluted as a tetramer followed by a smaller monomeric fraction (Fig. 1d), which is also the case with hTRPA1 and Δ1–688 hTRPA1[7,27]. The far UV CD spectra showed that Δ1–854 hTRPA1 has the characteristics of a predominately α-helical structure with minima at 208 and 222 nm

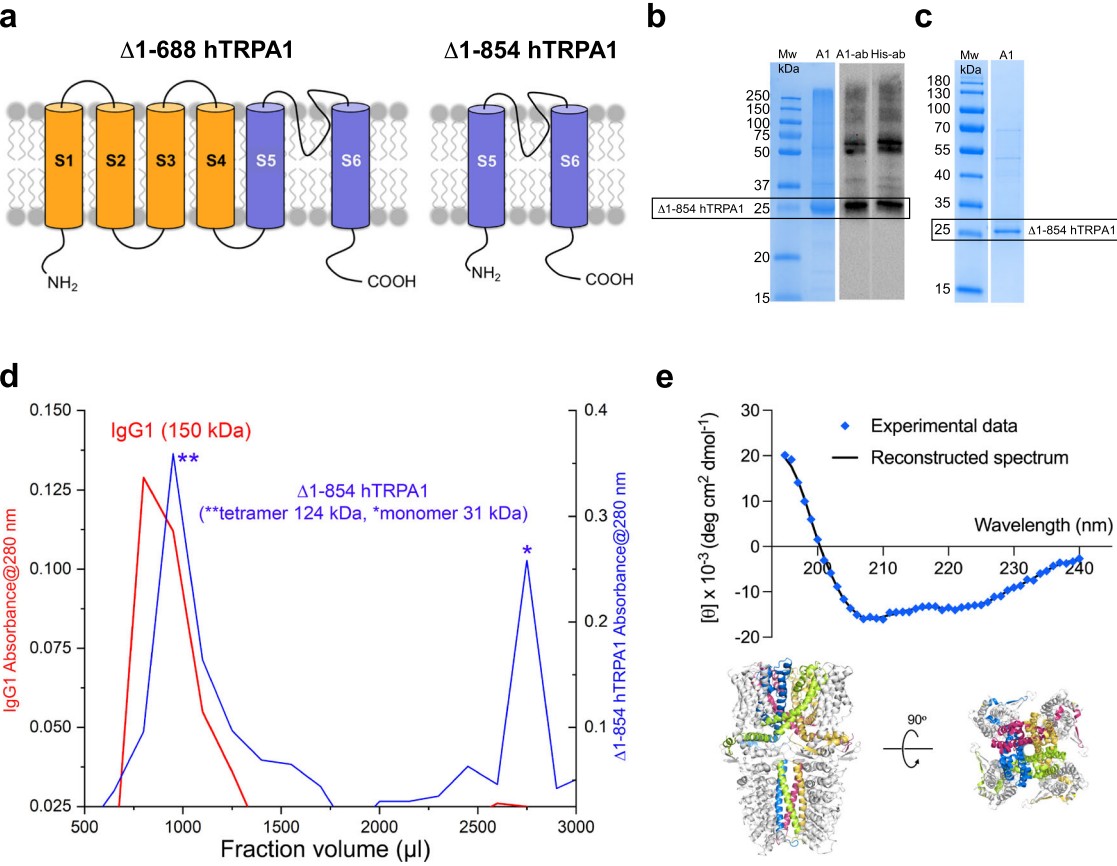

**Fig. 1 | Purified human TRPA1 without its N-terminal ankyrin repeat domain and transmembrane segments S1–S4. a–c** The N-ARD deleted hTRPA1 (Δ1–688 hTRPA1) and the further S1–S4 transmembrane domain deleted hTRPA1 (Δ1–854 hTRPA1) were purified and used in the present study. Δ1–688 hTRPA1 was purified from *Pichia pastoris* as previously described (ref. 7), whereas Δ1–854 hTRPA1 was purified from Hi-5 (**b**) and sf9 (**c**) insect cells (Mw, molecular weight). **b** Affinity-purified Δ1–854 hTRPA1 from Hi-5 cells is visualized by Coomassie staining (left gel, *n* = 2), and by Western blotting (right gel, *n* = 2) using either hTRPA1 antibody (left lane) or tetrahistidine antibody (right lane). **c** Affinity-purified Δ1–854 hTRPA1 from sf9 cells is visualized by Coomassie staining (*n* = 1). **d** As shown by the chromatogram, using a Sephadex 100 size exclusion column, Δ1–854 hTRPA1 eluted as a

tetramer (**) and monomer (*). Immunoglobulin with Mw of 150 kDa was used as a reference to indicate tetrameric fraction. **e** CD spectroscopy analysis (*n* = 25 replicate scans examined over four independent experiments; line shows fit-to-data from DichroWeb secondary structure analysis) disclosed typical characteristics of Δ1–854 hTRPA1 as a folded protein with similar content of α-helices, β-strands, turns and unordered residues when compared to hTRPA1 (PDB 6V9Y). The structure of Δ1–854 hTRPA1 is the colored part of the hTRPA1 cryo-EM structure (PDB 6V9Y) shown in a ribbon representation, from the side (left) and in a 90° rotational view of the pore domain of TRPA1 around the *x*-axis (right). Source data are provided as a Source Data file.

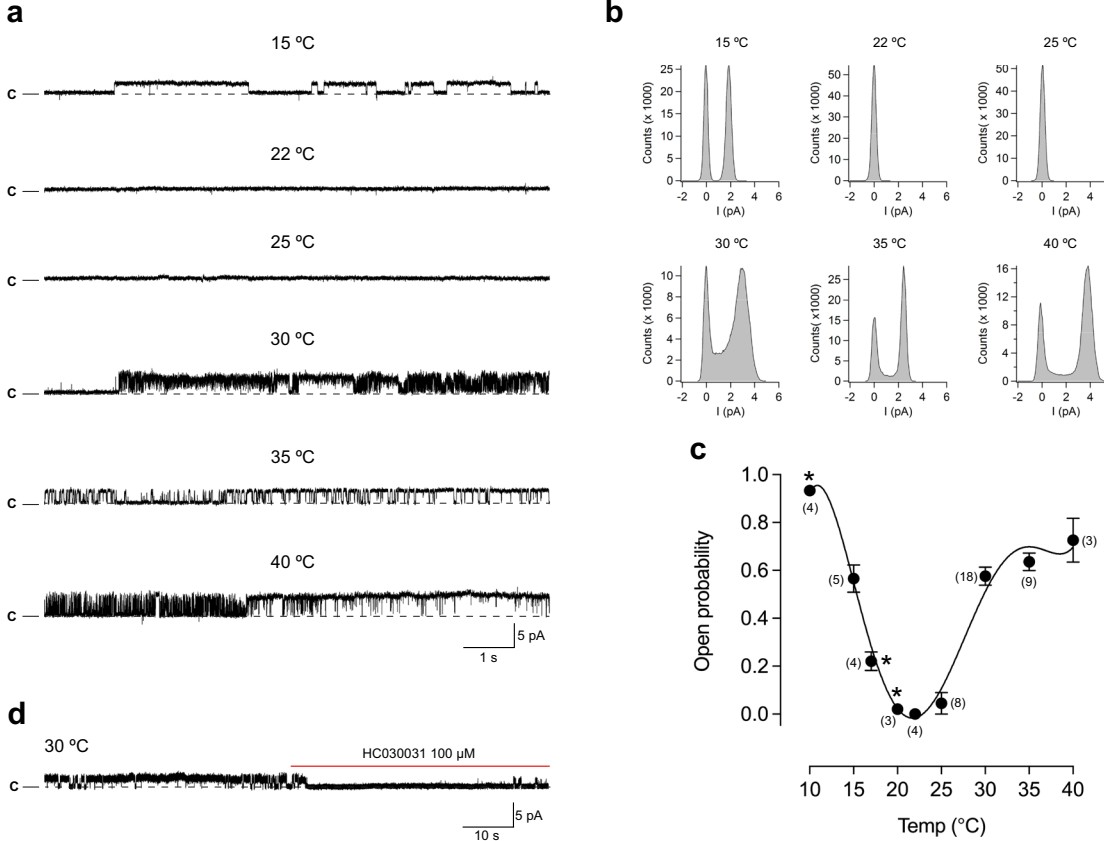

**Fig. 2 | The purified human TRPA1 without its N-terminal ankyrin repeat domain (Δ1–688 hTRPA1) is both cold- and heat sensitive.** Purified Δ1–688 hTRPA1 was reconstituted into planar lipid bilayers and single-channel currents were recorded with the patch-clamp technique in a symmetrical K$^+$ solution at a holding potential of +60 mV. **a, b** As shown by representative traces and corresponding histograms, exposure to various temperatures evoked outward single-channel currents. **c** The graph shows single-channel mean open probability ($P_o$) values as a function of different temperatures. Data are represented as the mean ± SEM of separate experiments (shown within parentheses). Data points marked with asterisk were published previously (ref. 7). **d** The selective TRPA1 antagonist HC030031 inhibited heat responses ($n = 3$). Black dashed line shows zero channel current level (c indicates the closed channel state) and upward deflections are channel openings. $Q_{10}$ cold = 0.018 (ref. 7); $Q_{10}$ heat = 29 ($T_1 = 20$ °C and $T_2 = 30$ °C, and corresponding $P_o$ values in Supplementary Table 1; Eq. (1)). Source data are provided as a Source Data file.

(Fig. 1e). The estimated secondary structure composition from CD spectroscopy reveals that Δ1–854 hTRPA1 contains 45% α-helices, 27% β-strands, 7% turns, and 21% unordered residues in solution. This content matches well with the 56% α-helices, 12% β-strands, 10% turns, and 22% unordered residues present in the Δ1–854 hTRPA1 part of the full hTRPA1 structure as determined by cryo-EM[28].

## The heat sensitivity is lost in Δ1–854 hTRPA1 but not in Δ1–688 hTRPA1

In agreement with the previous studies[7,11], Δ1–688 hTRPA1 was confirmed to be activated by cold (15 °C) when reconstituted into artificial planar lipid bilayers (Fig. 2, Supplementary Table 1). Here we show that Δ1–688 hTRPA1 also responded with pronounced channel activity to temperatures above 25 °C (Fig. 2, Supplementary Table 1). A near maximum single-channel open probability value was reached already at 30 °C with a calculated $Q_{10}$ value of 29, using $P_o$ values (Supplementary Table 1) at $T_1 = 20$ °C and $T_2 = 30$ °C [Eq. (1)].

The purified Δ1–854 hTRPA1, reconstituted into artificial planar lipid bilayers, responded with pronounced channel activity to temperatures below 20 °C (Fig. 3, Supplementary Table 1). Within the applied temperature interval of 20–15 °C, the single-channel open probability value reached a maximum close to 1 (Fig. 3c) with a calculated $Q_{10}$ value of 0.001, using $P_o$ values (Supplementary Table 1) at $T_1 = 20$ °C and $T_2 = 15$ °C [Eq. (1)]. Although the heat sensitivity within 25–40 °C was lost, Δ1–854 hTRPA1 still displayed activity with similar

single-channel open probability within this temperature interval (Fig. 3a–c). Single-channel kinetics were further analyzed using dwell-time histograms to determine the time constant ($\tau$) for the closed and open single-channel states at various temperatures (Supplementary Fig. 1, Supplementary Table 2). Only at 15 °C, the time constant for the closed channel state was lower than for the open state. At 15 °C, the time constant for the open state is 52–131-fold higher than at the other temperatures (Supplementary Table 2). The single-channel open probability values calculated from these experiments are in good agreement with those presented in Supplementary Table 1 and Fig 3c. The rather similar time constant for the closed state within 25–40 °C as well as the open state within 25–40 °C provide evidence that the channel pore opening within this temperature interval is temperature-independent.

The activity of Δ1–688 hTRPA1 and Δ1–854 hTRPA1 was abolished by the TRPA1 inhibitor HC030031 (Figs. 2d and 3d, e, Supplementary Fig. 3b), at concentrations that abolished purified hTRPA1 and Δ1–688 hTRPA1 single-channel activity evoked by chemical ligands, temperature and mechanical stimuli[7,11,29–31]. Furthermore, no channel activity was observed in membranes, without the purified hTRPA1 proteins in the presence of the detergent Fos-Choline-14, when exposed to the temperatures used in the present study[7,11,29–31].

It is of note that most of our recordings do not cover all temperatures in the very same experiment, which also seems to be the case in similar studies of purified TRPV1, TRPM8, and TRPM3[9,32,33].

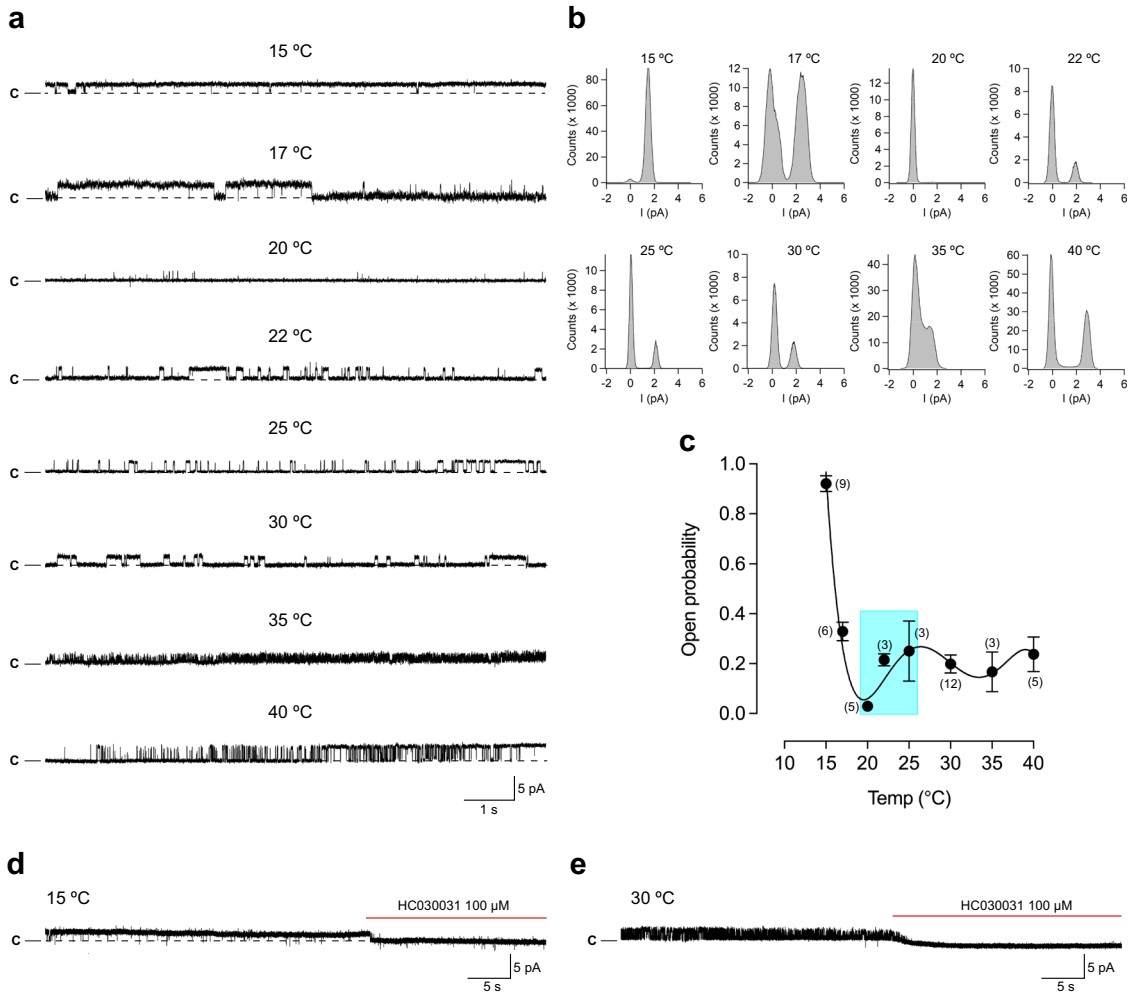

**Fig. 3 | The purified human TRPA1 without its N-terminal ankyrin repeat domain and transmembrane segments S1–S4 (Δ1–854 hTRPA1) is cold- but not heat sensitive.** Purified Δ1–854 hTRPA1 was reconstituted into planar lipid bilayers and single-channel currents were recorded with the patch-clamp technique in a symmetrical K⁺ solution at a holding potential of +60 mV. **a, b** As shown by representative traces and corresponding histograms, exposure to various temperatures evoked outward single-channel currents. **c** The graph shows single-channel mean open probability ($P_o$) values as a function of different temperatures.

Data are represented as the mean ± SEM of separate experiments (shown within parentheses). Colored area relates to the Δ1–854 hTRPA1 intrinsic tryptophan fluorescence measured within the same temperature interval (Fig. 6g). **d, e** The selective TRPA1 antagonist HC030031 inhibited both cold and heat responses (each $n = 4$). Black dashed line shows zero channel current level (c indicates the closed channel state) and upward deflections are channel openings. $Q_{10}$ cold = 0.001 ($T_1 = 20$ °C and $T_2 = 15$ °C, and corresponding $P_o$ values in Supplementary Table 1; Eq. (1). Source data are provided as a Source Data file.

Nevertheless, in a few experiments, we were able to capture channel activity within the whole temperature range in a single membrane patch (Supplementary Figs. 2 and 3). Regardless, at each temperature, the channel activity often consisted of various gating patterns within the same recording and between recordings (Figs. 2–5, Supplementary Figs. 2–4). This mixed channel behavior is consistent with our previous studies on purified hTRPA1 exposed to ligands, temperature, and pressure[7,11,29–31], and is also observed without explanation in bilayer single-channel recordings of purified TRPV1, TRPM8, and TRPM3[9,32,33]. Furthermore, we observed various single-channel conductance states at all temperatures (Figs. 4 and 5). A similar main conductance state could be identified at cold (15 and 17 °C) and warm temperatures (30–40 °C) for Δ1–688 hTRPA1 (55–65 pS) and Δ1–854 hTRPA1 (46–58 pS), as shown in Figs. 4 and 5. However, as shown by most traces, the various sub-conductance states seem to appear more often than the main conductance state, and at 15 °C channel openings mostly consisted of low sub-conductance states for both Δ1–688 hTRPA1 and Δ1–854 hTRPA1 (Figs. 2 and 3, Supplementary Figs. 2 and 3). A similar drop in single-channel conductance with cold has been observed for purified hTRPA1[7,11] as well as heterologously expressed mouse

TRPA1[34,35]. Clearly, future in-depth biophysical studies, as performed with TRPV1[36,37], are needed to dissect the complex gating mechanism of hTRPA1 in response to temperature, including interburst gaps (channel closure) and channel flickering behavior as well as single-channel conductance states. Nevertheless, it is the change in single-channel open probability and not single-channel conductance that explains the cold sensitivity of mouse TRPA1[34,35]. Likewise, in our study the hTRPA1 cold sensitivity is appropriately reflected by the analysis of single-channel open probability, allowing us to describe a unique relationship between structure and temperature-dependent activation of hTRPA1.

### Δ1–688 hTRPA1 and Δ1–854 hTRPA1 display different structural rearrangements in response to cold and heat
To study lipid bilayer-independent conformational changes in response to cold and heat, we exposed purified detergent-solubilized Δ1–688 hTRPA1 and Δ1–854 hTRPA1 to various temperatures and measured the intrinsic tryptophan fluorescence signal (Fig. 6). The maximum fluorescence signal was larger in Δ1–688 hTRPA1 compared to Δ1–854 hTRPA1 (Fig. 6), most likely because of more tryptophans in

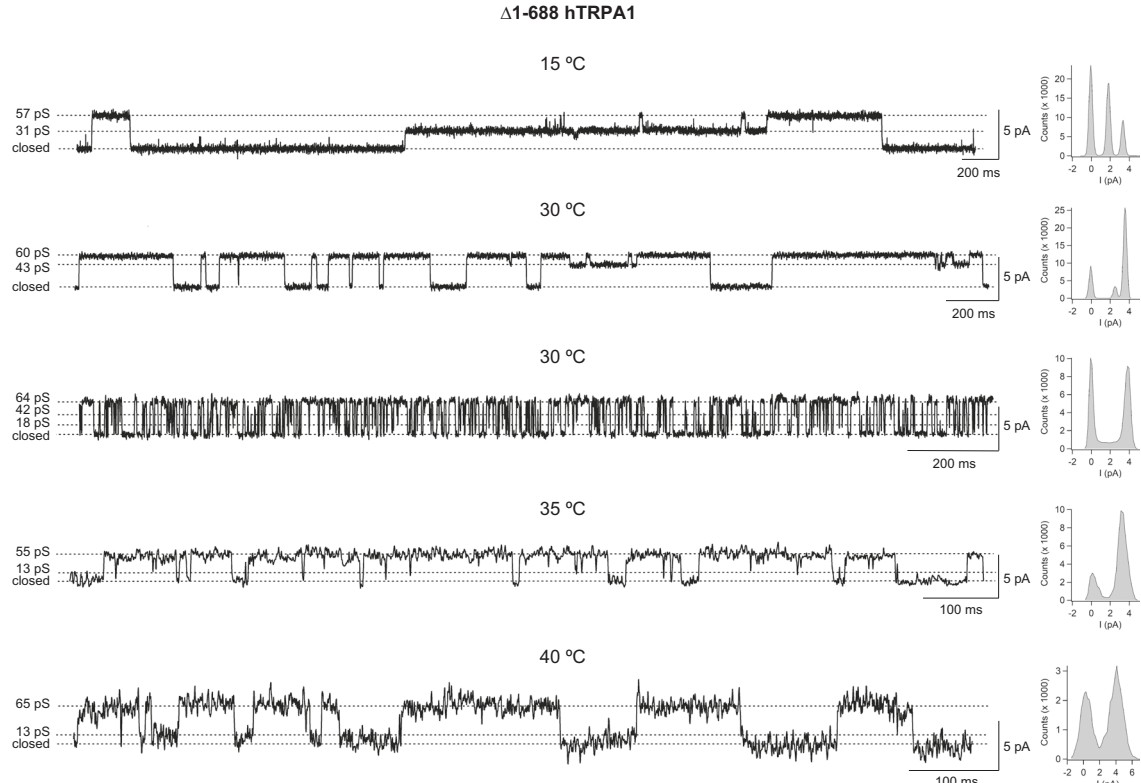

**Fig. 4 | The purified Δ1–688 hTRPA1 displays various single-channel conductance states.** As shown by representative traces and corresponding histograms, the full single-channel open state (main conductance) had a similar conductance (55–65 pS) at the temperatures investigated in the present study. In addition, several sub-conductance states (13–43 pS) were identified. The purified Δ1–688 hTRPA1 was reconstituted into planar lipid bilayers, and channel currents were recorded with the patch-clamp technique in a symmetrical K⁺ solution at a holding potential of +60 mV (upward deflection shows open-channel state).

Δ1–688 hTRPA1 (Fig. 7). Heat almost completely extinguished the Δ1–688 hTRPA1 fluorescence signal (Fig. 6b, f) and caused obvious quenching of the Δ1–854 hTRPA1 fluorescence signal (Fig. 6d, h). When lowering the temperature from 25 to 19 °C a sudden major drop of the Δ1–854 hTRPA1 fluorescence signal appeared, after which only a minor change of the protein fluorescence signal was observed with further cooling (Fig. 6c, g). A small gradual decrease in tryptophan fluorescence was also observed for Δ1–688 hTRPA1 when lowering the temperature from 22 °C (Fig. 6a, e). In the presence of carvacrol, cold caused a larger quenching of Δ1–688 hTRPA1 fluorescence (Fig. 6e). Notably, the hump at 305 nm in the Δ1–688 hTRPA1 heat fluorescence spectrum (Fig. 6b) is most likely fluorescence originating from tyrosines, and only visible because of the substantial quenching of the tryptophan fluorescence signal. The small but significant emission at 420 nm for Δ1–688 hTRPA1 (Fig. 6b) and possibly Δ1–854 hTRPA1 (Fig. 6c), could be related to fluorescence by tryptophans and/or tyrosines, since depending on their environment dramatic emission shifts to higher wavelengths can occur[38–40]. The spectra of the samples were restored back to the original value when recorded after a few hours at room temperature.

## TCEP inhibits Δ1–688 hTRPA1 and Δ1–854 hTRPA1 temperature-dependent channel activity

Both Δ1–688 hTRPA1 and Δ1–854 hTRPA1 contain cysteines that may become oxidized and form various intra- and intermolecular disulfide networks affecting channel function properties (Fig. 7). As shown previously, hTRPA1 cold- and heat responses and Δ1–688 hTRPA1 cold responses were prevented by the thiol reducing agent TCEP[11]. As shown in the present study, TCEP at the same concentration (1 mM) also completely prevented heat responses in Δ1–688 hTRPA1 as well as Δ1–854 hTRPA1 cold responses (Fig. 8). The effect of TCEP on Δ1–854

hTRPA1 heat responses was not investigated as its heat sensitivity was lost (Fig. 3).

## Cysteines 856, 1021, and 1025 modulate hTRPA1 cold activation

The role of cysteines 856, 1021, and 1025 in voltage-dependent cold responses was examined by measuring whole-cell currents in HEK293T cells expressing three single cysteine mutants (C856A, C1021A, C1025A) or one double cysteine mutant (C1021A/C1025A) of hTRPA1 (Fig. 9). Voltage steps from –160 to +200 mV were applied from a holding potential of 0 mV in extracellular solution and the conductance-to-voltage (G/V) relationships were compared at 25 and 15 °C (Fig. 9a–f). At 25 °C, the half-maximum activation voltage ($V_{50}$) of the C856A construct was significantly rightward shifted to $115 \pm 8$ mV ($n = 11$) compared with wild-type channels ($86 \pm 4$ mV; $n = 21$), indicating that the mutation modified the voltage-dependent gating (Fig. 9b, e). In wild-type channels, cooling to 15 °C significantly increased current transients at negative membrane potentials (see Fig. 9a, b; triangles) and significantly shifted $V_{50}$ to $58 \pm 5$ mV. In C856A, cooling did not produce the inward current transients and $V_{50}$ was shifted only to $95 \pm 8$ mV; Fig. 9a, b, f). On the other hand, the leftward shift of the G/V relationship upon cooling was significantly more pronounced in C1021A, C1025A, and C1021A/C1025A and, similar to the wild-type channels, these constructs exhibited transient inward currents upon hyperpolarization, suggesting a cold-induced increase in channel open probability at physiological voltages (Fig. 9a–d, f). To further explore the extent to which the three cysteines are involved in the cold-dependent activation of hTRPA1, we also performed steady-state experiments using a protocol similar to that used recently for TRPM8[41] (Fig. 9g). Thus, we measured cold-induced currents under conditions in which

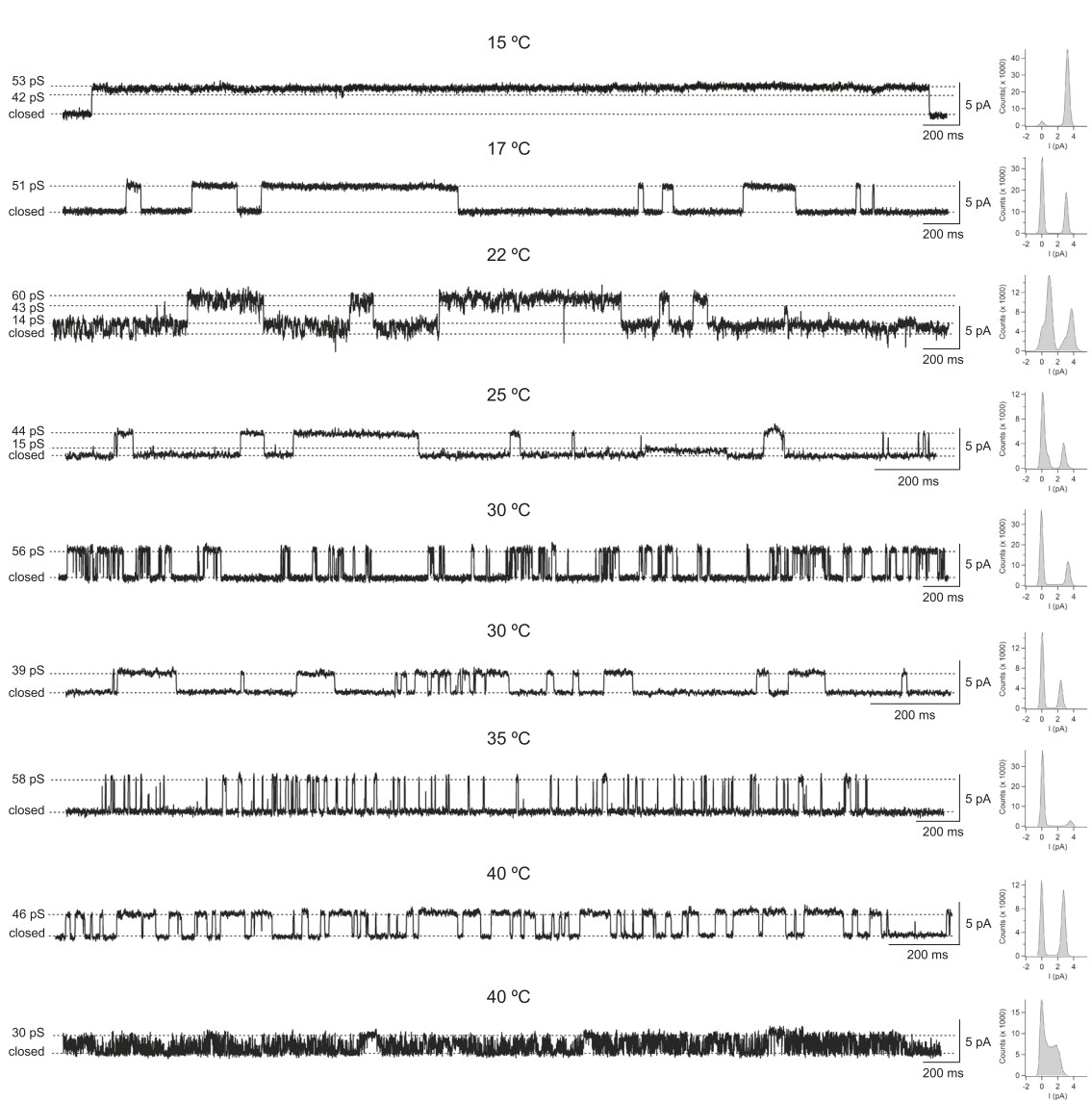

**Fig. 5 | The purified Δ1–854 hTRPA1 displays various single-channel conductance states.** As shown by representative traces and corresponding histograms, several conductance states of which the main state had a conductance of 44–60 pS were identified at the temperatures investigated in the present study. Sub-conductance states were in the range of 14–43 pS, as also evident in Fig. 3a and

Supplementary Fig. 3. The purified Δ1–854 hTRPA1 was reconstituted into planar lipid bilayers, and channel currents were recorded with the patch-clamp technique in a symmetrical K⁺ solution at a holding potential of +60 mV (upward deflection shows open-channel state).

the putative voltage sensor was supposed to be only partially activated (+80 mV) and the putative cold sensor strongly activated (5 °C). The responses were related to the currents produced by the presence of a saturating concentration of the non-covalent agonist of TRPA1 carvacrol (100 μM). Carvacrol-induced maximum current densities measured at +80 mV in C856A (197 ± 22 pA/pF; $n = 6$), C1021A (289 ± 56 pA/pF; $n = 5$), C1025A (196 ± 30 pA/pF; $n = 5$), and C1021A/C1025A (200 ± 35 pA/pF; $n = 5$) were not significantly different from wild-type channels (265 ± 37 pA/pF; $n = 7$; $P = 0.261$, one-way ANOVA). Whereas the maximum relative response of C856A to cold at +80 mV was not significantly different from wild-type channels (Fig. 9h), single mutation at C1021 and the double mutation C1021A/C1025A led to a significant decrease in relative response to cold at +80 mV. Together, the results suggest that the cysteines 856, 1021, and C1025 are involved in both the temperature- and voltage-dependent gating of hTRPA1, with a more general role of C856

in TRPA1 channel gating and a specific role of C1021 and C1025 in cold-dependent gating.

## Discussion

Several TRP channels are involved in mammalian thermosensation with some overlap in temperature responsiveness[1–5], but only a few thermoTRPs have been shown to intrinsically respond to a change in temperature[6,8–11]. In our studies on purified hTRPA1 and AgTRPA1, we found that hTRPA1 responded to cold and AgTRPA1 to heat and that these intrinsic temperature responses remained in the absence of the N-ARD and part of pre-S1 region[7,8]. We speculated that the redox state and/or other post-translational modifications controlled the hTRPA1 intrinsic temperature sensitivity also allowing hTRPA1 to be activated by heat[11], which would support findings indicating involvement of TRPA1 in mammalian heat detection in vivo[42–47]. Indeed, cold- and heat-induced activity of hTRPA1 was found to be dependent on the channel redox state, and both oxidants and ligand activators, such as hydrogen

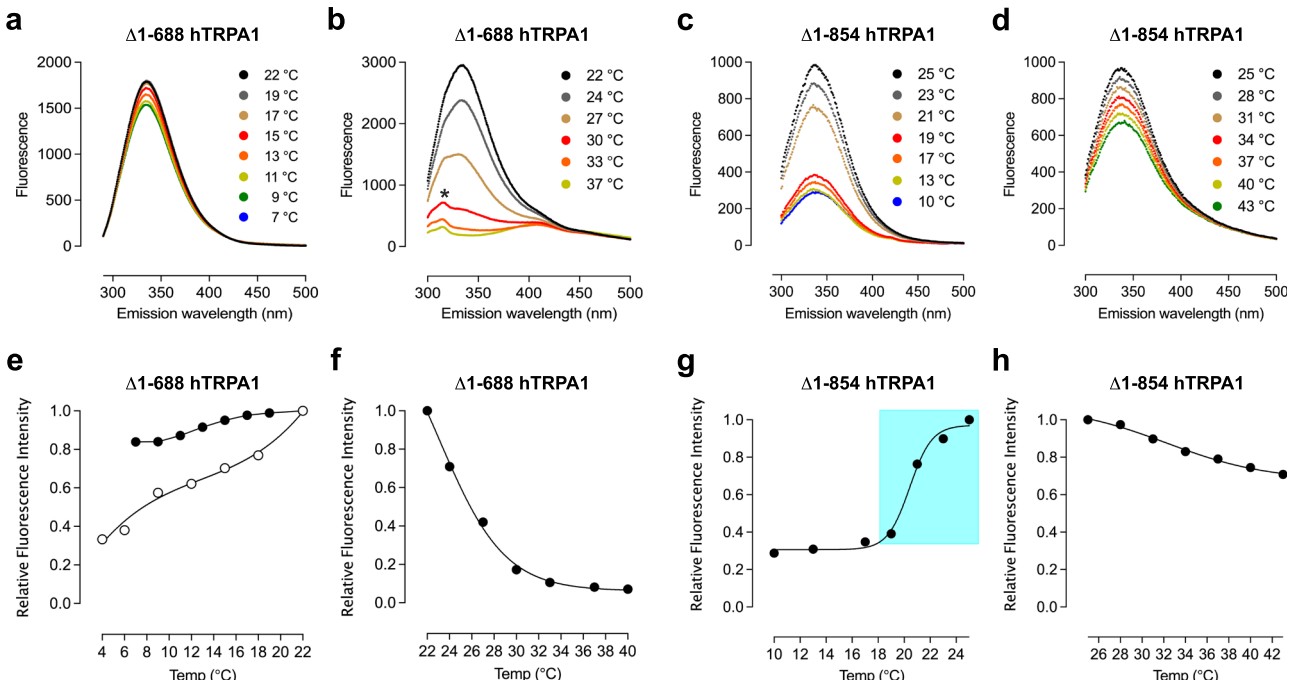

**Fig. 6 | Cold and heat cause lipid bilayer-independent conformational changes of purified Δ1–688 hTRPA1 and Δ1–854 hTRPA1. a–d** Spectra show different conformational changes induced by cold or heat for Δ1–688 hTRPA1 (**a**, **b**) and Δ1–854 hTRPA1 (**c**, **d**). **b** The hump (*) was only observed for Δ1–688 hTRPA1 and above 22 °C. **e–h** The intrinsic tryptophan fluorescence intensity, emitted at 335 nm, for each indicated temperature was related to that of 22 °C (**a**, **b** and **e**, **f**) or 25 °C (**c**, **d** and **g**, **h**) and expressed as relative fluorescence intensity in the graphs.

**g** Colored area relates to the Δ1–854 hTRPA1 channel open probability measured within the same temperature interval (Fig. 3c). **e** Shown is also the effect of carvacrol (100 μM, open circles) on Δ1–688 hTRPA1 cold responses. At 22 °C, carvacrol itself emitted fluorescence that was subtracted when its effect on cold was determined. Data are represented as the mean ± SEM of three independent experiments. Source data are provided as a Source Data file.

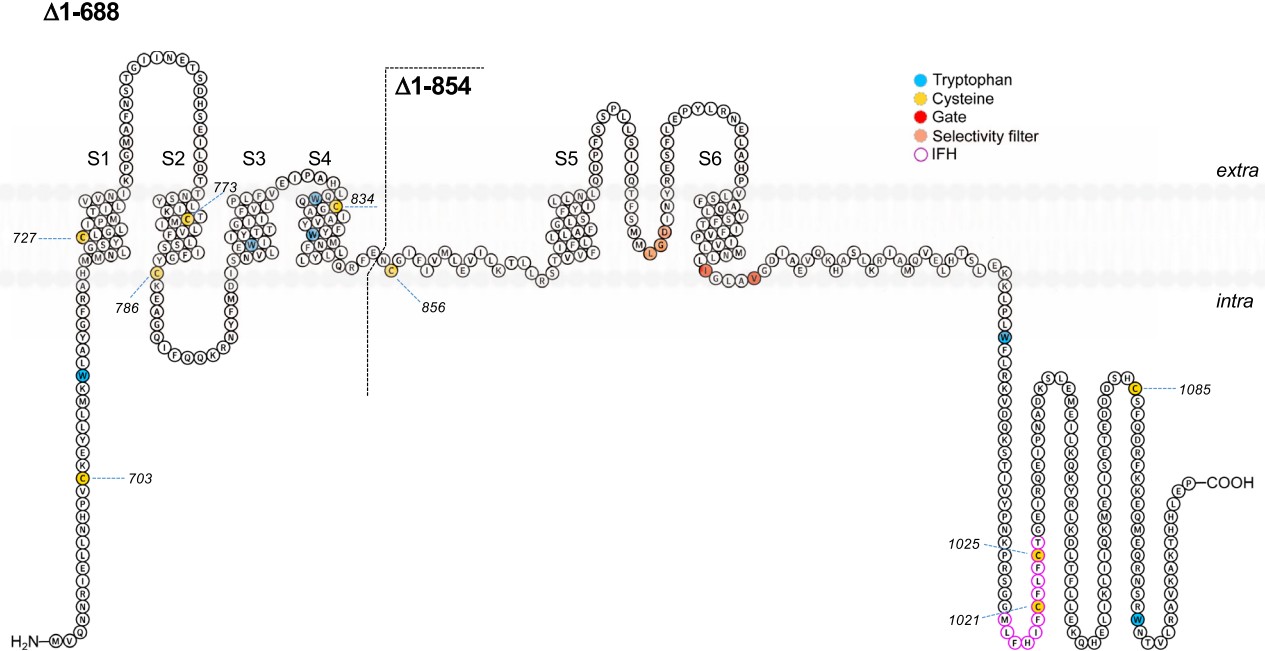

**Fig. 7 | Topology plot of Δ1–688 hTRPA1 and Δ1–854 hTRPA1.** Shown is hTRPA1 without its N-ARD and part of the pre-S1 region (Δ1–688 hTRPA1) and its further truncated isoform (Δ1–854 hTRPA1). Highlighted are tryptophans (blue circle) with the capacity to contribute to the intrinsic fluorescence recorded in Fig. 6, and

cysteines (yellow circle) of which the role of C856, C1021, and C1025 in hTRPA1 cold responses was explored in whole-cell patch-clamp studies (Fig. 9). C1021 and C1025 are located within a short helix, termed interfacial helix (IFH), that makes close contact with the voltage sensing-like domain (VSLD) consisting of S1–S4.

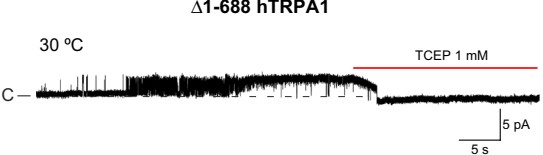

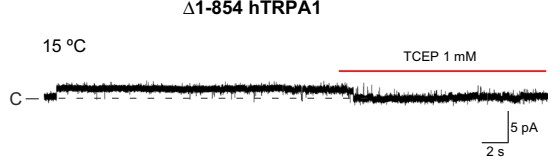

**Fig. 8 | Δ1−688 hTRPA1 and Δ1−854 hTRPA1 heat and cold sensitivity are dependent on the channel redox state.** Purified Δ1−688 hTRPA1 and Δ1−854 hTRPA1 were reconstituted into planar lipid bilayers and single-channel currents were recorded with the patch-clamp technique in a symmetrical K⁺ solution at a holding potential of +60 mV. As shown by traces, the thiol reducing agent TCEP eliminated the channel activity of Δ1−688 hTRPA1 evoked by heat ($n = 6$) and Δ1−854 hTRPA1 evoked by cold ($n = 5$). Black dashed line shows zero channel current level (c indicates the closed channel state) and upward deflections are channel openings.

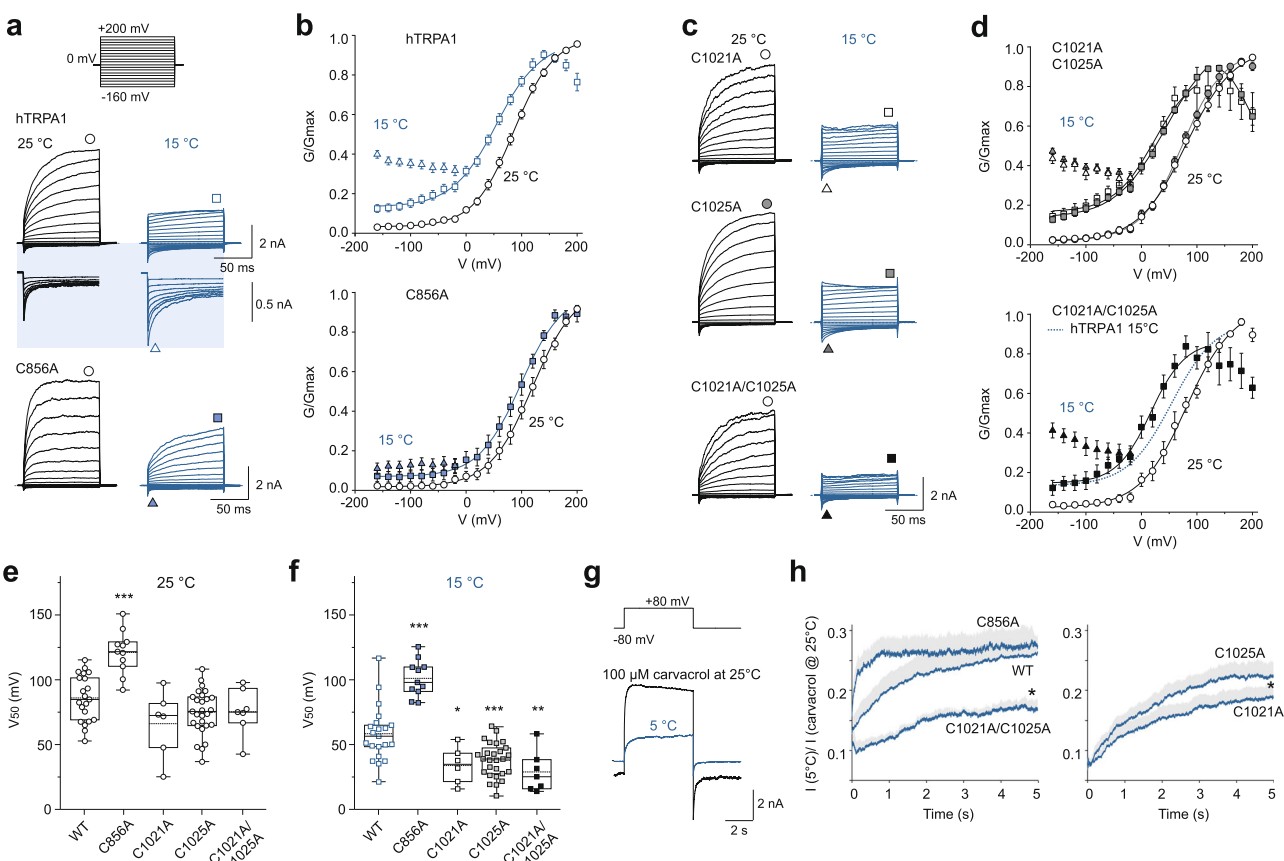

**Fig. 9 | The voltage-dependent activation properties of cysteine mutants of human TRPA1, C856A, C1021A, C1025A, and C1021A/C1025A, are differentially modulated by cold. a** Representative time course of whole-cell current amplitudes measured from HEK293T cells expressing hTRPA1 (wild-type or C856A) channels. Voltage step protocol (steps from −160 to +200 mV, steps +20 mV) is shown in the inset. Zoomed-in view of inward current traces is highlighted in blue. Conductances plotted in **b** are taken at times marked by circles and squares for steady-state, and by triangles for peak inward transient currents. **b** Average conductance-to-voltage (G/V) relations obtained at 25 °C (open black symbols) and at 15 °C (open blue squares) from wild-type human TRPA1 (hTRPA1; $n = 21$) and from the C856A construct ($n = 11$), normalized to maximum conductance $G_{max}$ for each cell (values with error bars are expressed as mean ± SEM). The relationships were fitted with a Boltzmann equation. **c, d** Average G/V relations (values with error bars are expressed as mean ± SEM) obtained at 25 °C and at 15 °C for the C1021A ($n = 6$), C1025A ($n = 27$), and C1021A/C1025A ($n = 7$) constructs. The blue dotted line indicates the average G/V relation obtained at 15 °C for wild-type channels. **e, f** Box plots summarizing the values of the half-maximum activation voltage ($V_{50}$) of the wild-

type and cysteine mutants of hTRPA1 obtained at 25 °C (**e**) and at 15 °C (**f**). In the box plots, the horizontal lines indicate the median (solid line) and mean (dashed line), the bottom and top edges of the box indicate the interquartile range, and the whiskers represent the maximum and minimum data points. *$P < 0.05$, **$P < 0.01$ and ***$P < 0.001$ indicate statistically significant differences compared to wild-type TRPA1 using the one-way ANOVA and Dunnett's post hoc comparison tests. **g** Representative time course of superimposed whole-cell current responses measured from a HEK293T cell expressing human hTRPA1 channels in control bath solution at 5 °C and then in the presence of 100 μM carvacrol at 25 °C. The voltage step protocol (step from −80 to +80 mV) is shown above. **h** Average time course of responses from wild-type and mutant TRPA1 channels obtained by using the voltage protocol shown in **g**. Colored lines and envelopes are the mean ± SEM obtained from 7 (wild-type), 6 (C856A), 5 (C1021A), 5 (C1025A), and 5 (C1021A/C1025A) independent measurements. Relative current responses measured at the end of the trace (*) in C1021A/C1025A and C1021A were significantly different from wild-type channels ($P = 0.004$ and $P = 0.017$, respectively; unpaired one-sided Student's $t$-test). Source data are provided as a Source Data file.

peroxide (H₂O₂), allyl isothiocyanate (AITC), acrolein, and carvacrol-modified hTRPA1 activation by temperature in intact HEK293 cells and mouse trachea sensory neurons[11]. We, therefore, concluded that the heat-sensing property of TRPA1 is conserved in mammalians, in which

TRPA1 may contribute to sensing warmth and uncomfortable heat in addition to noxious cold[11]. Similar conclusions regarding TRPA1 as a heat sensor of physiological relevance have been reached in recent publications[24,48–50].

In this study, based on single-channel open probability analysis we show that hTRPA1 without its N-ARD (Δ1−688 hTRPA1) retains its bidirectional thermosensitive profile but with increased thermosensitivity as indicated by $Q_{10}$ values of 0.025 (hTRPA1) and 0.018 (Δ1−688 hTRPA1) for cold[7], and 6 (hTRPA1)[11] and 29 (Δ1−688 hTRPA1) for heat. Likewise, the heat-sensitivity of purified Ag-TRPA1(A) without its N-ARD (Δ1−776 AgTRPA1) was estimated higher than that of Ag-TRPA1(A) for which a $Q_{10}$ value of 28 could be calculated[8]. This suggests a modulatory role of the N-ARD in TRPA1 thermosensation, but may not necessarily be driven by temperature-sensitive domains within N-ARD. This is similar to what has been proposed for the chemo- and mechanosensitivity of purified hTRPA1[7,30,31]. Further deletion of VSLD in hTRPA1 (Δ1−854 hTRPA1), keeping most of the S4−S5 linker with the redox-sensitive C856[25] and N855, the mutation of which in humans causes stress-induced cold hypersensitivity[51], created a channel with a dramatic increase in cold sensitivity and lost heat sensitivity within 25−40 °C. The latter may indicate that heat-sensitive structures are within VSLD or that the heat sensitivity is located in CTD and that the VSLD is supporting CTD in heat-evoked channel gating. The increase in cold sensitivity with further truncation, as demonstrated by a $Q_{10}$ value of 0.001 for Δ1−854 hTRPA1, suggests that also VSLD negatively modulates the cold-induced responses, which may be triggered by thermosensor loci within both S5−S6 and CTD. Interestingly, this brake of cold responses can be counteracted by both electrophilic (AITC and acrolein) and non-electrophilic (carvacrol) TRPA1 activators, interacting with binding sites within the N-ARD and VSLD[7,27,48,52,53], as shown by their ability to sensitize mammalian TRPA1 cold responses[11,24,48,54]. The selective TRPA1 antagonist HC030031 completely prevented cold and heat responses of hTRPA1 and its truncated isoforms in previous and present studies[7,11] supporting the idea that HC030031 inhibits hTRPA1 by interacting with N855 and the CTD[55].

Some studies on mammalian TRPA1 cold responses provide a $Q_{10}$ value ~0.1[34,35,56]. The reason why these studies and the present study record cold responses for TRPA1 heterologously expressed in mammalian cell lines with $Q_{10}$ values that are far from those obtained for the purified hTRPA1s in bilayer electrophysiology recordings is not obvious, but the TRPA1 redox environment and other cellular environmental factors including temperature-sensitive potassium channels most likely influence TRPA1 activity[52,53]. Furthermore, it has been shown that phospholipids influence the temperature responses of TRPV1 and TRPM8 in bilayer recordings[6,9,32]. In addition, it is possible that the reconstituted channel does not adopt its native state completely. Together, this could explain why $Q_{10}$ values for not only TRPA1 but also TRPV1, TRPM8, and TRPM3 in a cellular environment not always are in good agreement with $Q_{10}$ values obtained for these channels when purified and reconstituted into artificial lipid bilayers[9,32,33,57]. In the present study, due to the failure of heterologous cellular expression of N-ARD-truncated hTRPA1s[58,59], we cannot explore whether $Q_{10}$ values for purified hTRPA1s can be obtained in whole-cell recordings. However, by comparing the $Q_{10}$ values obtained for purified hTRPA1s, we can clearly demonstrate the relationship between the structure and activity of hTRPA1 in response to cold and heat.

Because hTRPA1 is an inherent mechanosensitive ion channel gated by force-from-lipids[31], it could be argued that hTRPA1 is indirectly thermosensitive by responding to temperature-dependent bilayer fluidity changes. However, the lipid 1,2-diphytanoyl-sn-glycero-3-phosphocholine form bilayers that are structurally stable within the temperature interval studied here and in the previous studies[60], and in which no currents were detected in the absence of hTRPA1 when exposed to the same test temperatures[7,8,11]. Further evidence of hTRPA1 being intrinsically thermosensitive is provided by lipid bilayer-independent measurements of the intrinsic tryptophan fluorescence activity of hTRPA1[11], and its truncated isoforms in the present study as further discussed below.

The intrinsic tryptophan fluorescence spectroscopy measurements, in contrast to patch-clamp recordings, can detect structural rearrangements of TRP channels associated with ligand interaction and temperature changes[11,61]. The stronger tryptophan fluorescence signal in Δ1−688 hTRPA1 compared to Δ1−854 hTRPA1 indicates a major contribution from the tryptophans located within S3−S4 and the pre-S1 region that may mask the detection of any CTD conformational changes, signaled by only its two tryptophans, in response to heat and cold. The extensive heat-induced quenching of the Δ1−688 hTRPA1 tryptophan fluorescence and the characteristic hump, which is most likely signaled by tyrosines, in both Δ1−688 hTRPA1 and hTRPA1[11] but not in Δ1−854 hTRPA1 fluorescence spectra, suggest that the heat temperature sensor is located within the VSLD. However, structural rearrangements of VSLD could possibly also occur as a result of the heat-induced conformational changes of the CTD as revealed by Δ1−854 hTRPA1. Thus, it could be that heat is properly detected by CTD, but without coupling to the VSLD heat cannot gradually be transformed into gating of the channel, as shown by the lost heat-sensitivity in Δ1−854 hTRPA1 within 25−40 °C. Notably, some conformational changes of the pore region must still occur in the absence of VSLD within 25−40 °C, but independent of changes in temperature as the single-channel open probability was similar within this temperature range. Importantly, the Δ1−854 hTRPA1 channel activity at 30 °C was abolished by HC030031 and its single-channel conductance within 25−40 °C was in the same range as for Δ1−688 hTRPA1 in the present study and hTRPA1[11], indicating that uncoupling of CTD and VSLD affected mainly the heat-sensitivity and not the pore function of Δ1−854 hTRPA1. Thus, it could be that a heat sensor is located in either CTD or VSLD, or present in both CTD and VSLD possibly acting in concert to gate hTRPA1. The cold-induced tryptophan fluorescence changes of Δ1−688 hTRPA1 were modest and could be the result of minor rearrangement of the VSLD, sensing cold directly or indirectly by cold-evoked structural changes of S5−S6, which lacks tryptophans. Furthermore, the small cold-evoked structural rearrangement of the CTD, as shown for Δ1−854 hTRPA1 when lowering the temperature from around 20 °C, may also take part in Δ1−688, but is not visible because of the strong tryptophan signal from VSLD. Importantly, tryptophan fluorescence measurements of Δ1−854 hTRPA1 revealed a sudden drop in tryptophan fluorescence within 25−20 °C indicating an intermediate state between heat and cold conformations of the CTD, and thus it is proposed that the CTD contains a bidirectional temperature switch priming hTRPA1 for either cold or heat. This fits well with the lowest single-channel open probability obtained at 20−22 °C for hTRPA1[11] and its truncated isoforms in the present study. This raises the possibility that cold- and heat-induced conformational changes of CTD, which in hTRPA1 is in close contact with the pre-S1 region and the S4−S5 linker connecting the VSLD and pore domain, is a key mechanism in the temperature gating of hTRPA1. Interestingly, swapping the CTD between TRPV1 and TRPM8 caused opposite temperature sensitivity[12], and it was suggested that a folded-unfolded transition of a specialized temperature-sensitive structure in the CTD of TRPM8 is associated with an increase in the heat molar capacity required for its gating by cold[16]. However, a recent study suggested that TRPM8 cold-sensitivity is located in specific regions of the N-terminal cytoplasmic domain[62]. It should be noted that the comparison of Δ1−688 hTRPA1 and Δ1−854 hTRPA1 fluorescence signaling may be complicated, since the two remaining tryptophans in Δ1−854 hTRPA1 that are located within the CTD may experience a different local environment in the absence of VSLD. Nevertheless, our results clearly show that cold and heat responses within the same construct involve different channel conformational changes. Also, there is a substantial rearrangement of the CTD at the interface between warm and cold temperatures as revealed by studies of Δ1−854 hTRPA1.

As mentioned earlier, electrophiles and carvacrol sensitized the cold-induced hTRPA1 whole-cell currents studied in

HEK293T cells[11,24,48,54]. Furthermore, the cold-evoked hTRPA1 conformational change as measured by tryptophan fluorescence signaling was much larger in the presence of carvacrol[11]. Whereas carvacrol most likely binds within the VSLD[48], electrophilic TRPA1 activators are supposed to trigger responses by interacting with highly reactive cysteines (Cys621, Cys641, and Cys665) in the hTRPA1 N-ARD/pre-S1 region[28,63,64]. However, electrophiles such as N-methyl maleimide (NMM), (E)-2-alkenals, AITC, cinnamaldehyde, and *p*-benzoquinone can activate TRPA1 by interacting with other cysteines and lysines outside the N-ARD[7,27,28,65,66]. Interestingly, as determined by mass spectrometry, the NMM labeling of C834 in S4 decreased with increasing concentrations of NMM in both hTRPA1 and Δ1–688 hTRPA1, indicating a structural rearrangement of S4 by electrophiles[27] as confirmed by cryo-EM[28,64]. In the present study, we further examined the effect of carvacrol on the intrinsic tryptophan fluorescence activity in Δ1–688 hTRPA1 in response to cooling. In the presence of carvacrol, the tryptophan fluorescence was extensively quenched by cold temperatures which otherwise only produced a minor change in tryptophan fluorescence intensity. Taken together, these findings support a ligand-induced removal of a "cold brake" in the VSLD (Fig. 10).

The purified hTRPA1 is partially oxidized and thiol-modifying agents can either inhibit or potentiate its temperature responses[11]. In the present study, we found that Δ1–688 hTRPA1 heat-induced activity was abolished by the thiol-reducing agent TCEP, which also inhibited the cold-induced activity of Δ1–688 hTRPA1 as well as both cold and heat responses of hTRPA1[11]. Likewise, TCEP prevented cold-evoked Δ1–854 hTRPA1 channel activity in the present study. Taken together, these findings show that the redox state of hTRPA1 is a critical factor in determining its temperature sensitivity. The number of cysteines is substantially reduced with truncation and as a consequence, the intra/intermolecular disulfide network is most likely different in the hTRPA1 and its truncated isoforms. However, we reasoned that the cysteines 856, 1021, and 1025 are still important in coordinating CTD coupling to the pore region of Δ1–854 hTRPA1 as well as to the VSLD and pre-S1 region in Δ1–688 hTRPA1 and hTRPA1[48]. We, therefore, decided to further explore the role of these

cysteines in cold-evoked TRPA1 activation by expressing hTRPA1 in HEK293T cells for patch-clamp studies.

Our whole-cell patch-clamp studies on mutant channels show that cysteines 856, 1021, and 1025 are involved in the cold-dependent activation of hTRPA1. In particular, C856A was not affected at all by cold at negative membrane potentials, whereas the wild-type channels were potentiated by about two-fold. This residue has previously been recognized as one of the main targets of $O_2$ in hyperoxia[25], but it is also critical for the gating equilibrium and/or voltage-dependent activation of the channel[48]. The important structural role of this residue is clearly supported by recently resolved structures of hTRPA1 in different conformational states[19,28,64]. C856 is located at the N-terminal portion of the S4–S5 linker, which is directly involved in conformational changes leading to channel opening by covalent agonists. Interestingly, in the closed conformation under the apo conditions (PDB 6V9W), C856 contacts V875 in the fifth transmembrane domain S5 of the neighboring subunit. In contrast, the distance between C856 and V875 is more than 15 Å in a conformation opened by the irreversible electrophilic agonist iodoacetamide (PDB 6V9X). The valine 875 has been identified as a molecular determinant underlying the species-specific differences in the cold sensitivity of TRPA1[67]. The authors identified V875 in primate TRPA1, corresponding to G878 in rodents, and demonstrated that mutation G878V abolishes cold activation of the rat and mouse TRPA1. The role of this residue in cold-dependent gating was further supported by the analysis of the kinetic properties of the mutants of human TRPA1-V875G and mouse TRPA1-G878V at 12, 25, and 35 °C[24]. The other two cysteines, C1021 and C1025, are positioned within a short helix at the cytosol–membrane interface located near the S1 and S4 of the VSLD[64]. A role for this structural motif (termed the interfacial helix, IFH) has been suggested in voltage-dependent gating and phospholipid regulation[24,68]. In the structure of TRPA1 captured in complex with the reversible covalent agonist benzyl isothiocyanate (PDB 6PQP), the backbone carbonyl of C1025 directly interacts with N855, a residue preceding C856 and the site of a gain-of-function disease mutation causing syndrome characterized by pain that is triggered by physical stress, including noxious cold[51]. The

**Δ1-854 hTRPA1**

*Deletion*

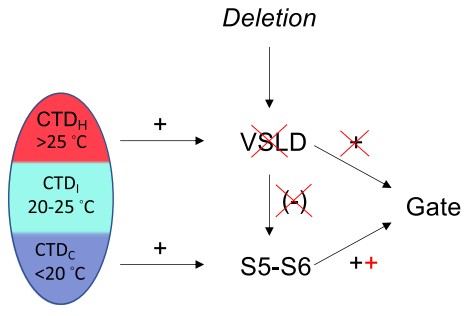

**Δ1-688 hTRPA1**

*Ligands and voltage*

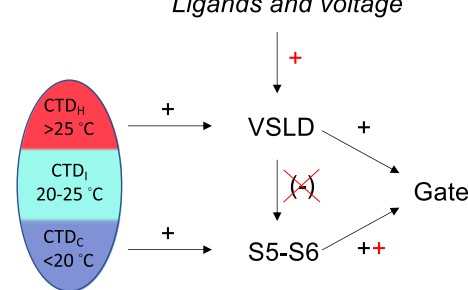

**Fig. 10 | Proposed mechanisms involved in hTRPA1 temperature detection.** The hTRPA1 intracellular C-terminal domain (CTD) contains cold ($CTD_C$) and heat ($CTD_H$) sensitive domains, of which the $CTD_C$ and S5-S6 transform cold exposure into channel opening whereas $CTD_H$ together with the voltage-sensing-like domain (VSLD) are involved in heat responses. The intermediate state ($CTD_I$), under which CTD undergoes sudden major structural rearrangement, as could be detected in the tryptophan fluorescence spectroscopy measurements of Δ1–854 hTRPA1 (containing only 2 tryptophans that are located within CTD), the temperature sensitivity of the channel is switched between heat and cold. When entering into the $CTD_C$ state, further cooling opens the channel fully within a narrow temperature range of 5–10 °C, but with minor changes in Δ1–854 hTRPA1 tryptophan fluorescence, suggesting that the CTD is in a stabilized state and conformational changes of S5-S6 occurs to open the gate. The same scenario seems valid for Δ1–688 hTRPA1 cold sensitivity, which also responded to cold with minor changes in tryptophan fluorescence in spite of additional 4 tryptophans of which 3 are within

S3–S4 and 1 in pre-S1 (Fig. 7). In contrast to cold, heat caused substantial Δ1–854 hTRPA1 CTD structural rearrangements at temperatures above 25 °C but without any increase in channel open probability, indicating that its heat sensitivity is not properly transmitted to the gate possibly by uncoupling of $CTD_H$ and VSLD. Indeed, maintained heat sensitivity is associated with the complete quenching of the tryptophan fluorescence in Δ1–688 hTRPA1 indicating a major structural rearrangement of S1–S4, and supports that coupling between $CTD_H$ and VSLD is crucial for proper heat activation of hTRPA1. The higher cold sensitivity of Δ1–854 hTRPA1 ($Q_{10} = 0.001$) compared to Δ1–688 hTRPA1 ($Q_{10} = 0.018$) suggests that the VSLD restricts cold-induced channel conformations. Interestingly, the non-electrophilic TRPA1 activator carvacrol and voltage may remove this inhibitory effect by interacting with the VSLD. This may also be the case for oxidants and electrophilic compounds such as AITC, acrolein, and NMM. Thus, the temperature sensitivity of TRPA1 and other thermoTRPs may be highly dependent on the channel redox state as well as other post-translational channel modifications.

double mutant C1021A/C1025A exhibited the most impaired cold responses, and thus it is possible that the formation of a disulfide bond between C1021 and C1025 has a key impact on the C1025–N855 interaction and channel gating. Together, this suggests the involvement of all three cysteines 856, 1021, and 1025 in the voltage- and cold-dependent gating of hTRPA1. This further supports our findings that the redox state of hTRPA1 is important in determining its temperature-sensitive properties.

Two recent studies using cryo-EM as the core technique to explain thermoTRP gating neither supported nor dismissed the molar heat capacity model or an allosteric coupling model determined by specific amino acids by which TRPV1 and TRPV3 are gated by heat[10,23]. As proposed for TRPV1, both large global conformational changes of multiple subdomains followed by small amino acid-specific outer pore conformational changes in a stepwise action are responsible for heat-dependent TRPV1 gate opening when capsaicin is bound in the vanilloid binding pocket that is formed by residues in the S2–S3 loop, S3, S4 and S4–S5 linker of one subunit, and S5 and S6 of the neighboring subunit[23]. Likewise, the heat-induced opening of TRPV3 is caused by a conformational wave involving the S2–S3 linker and the N and C-termini, and as a result modulation of the lipid occupying vanilloid-binding site[10]. Lipid displacement from the vanilloid pocket has also been proposed to cause heat activation of TRPV1[18], and cholesterol binding to mouse TRPA1 influences its functional response to AITC[69]. Interestingly, replacement with a single human amino acid in the pore region of drosophila TRPA1 reversed its thermosensitivity from heat to cold[70], and several VSLD lipid-binding pockets, also likely to bind carvacrol, are in close connection with CTD and N-ARD/pre-S1 region of hTRPA1[48]. Thus, in the case of hTRPA1, it will be interesting to further explore by cryo-EM if specific amino acids in the pore region are involved in cold-induced structural rearrangement and if any of the lipid-binding pockets in the VSLD are key for its heat activation. Naturally, the effect of redox on temperature-driven conformational changes and gating of hTRPA1 will also be of importance to study combining cryo-EM and mass spectrometry.

Our observation that point mutations at C856, C1021, and C1025 affect both voltage- and cold-sensitivity may fit with an allosteric model in which there is only one heat-activated temperature sensor[15]. This model is based on the presumption that independent heat- and voltage sensors exist that are coupled to the channel gate and to each other. The heat sensor is coupled to channel gating by a temperature-dependent allosteric coupling factor whose enthalpic contribution solely determines whether the channel is activated by both cooling and heating or only by cooling (Fig. 5 in ref. [15]). We demonstrate that the deletion of the VSLD (Δ1–854 hTRPA1) led to a selective loss of the heat-activated branch of the relationship between the single-channel open probability and temperature. This could support the allosteric coupling model over the heat capacity model[14], as the latter predicts that the steepness of the cold- and heat-activated branches of the single-channel open probability–temperature relationship is symmetric. Even though a single temperature sensor, e.g., located within the CTD would trigger both cold and heat responses, separate cold and heat pathways to the channel gate must exist as both functional bilayer experiments and tryptophan fluorescence studies of the purified hTRPA1s clearly showed that the VSLD is only needed for the heat sensitivity. However, it could also be that separate specialized temperature sensor domains in the CTD, the conformational transformation of which may involve protein folding–unfolding, dictate the allosteric coupling of CTD to the gate (Fig. 10). A similar role for CTD has been proposed in the cold activation of TRPM8[16], and may very well be involved in the temperature gating of TRPV1 and TRPV3[10,12,23].

The bidirectional temperature sensitivity of hTRPA1 may be a general property of all intrinsic wild-type thermoTRPs, as proposed by Clapham and Miller[14], but remains to be demonstrated. In this context, we did not find any activity of lipid bilayer reconstituted purified

AgTRPA1(A) when lowering the temperature to 15 °C[8]. However, AgTRPA1(A) displayed basal activity even at 15 °C when expressed in a cellular system[71]. In comparison with AgTRPA1(B), AgTRPA1(A) has additional amino acids at the distal end of the N terminus reducing its heat sensitivity[71], and perhaps any cold sensitivity. Thus, it is possible that post-translational modifications as well as lower temperatures are needed to reveal any intrinsic AgTRPA1 cold sensitivity. Although TRPA1 stands out among thermoTRPs with regard to the many cysteines (28 in hTRPA1) and extreme sensitivity to electrophiles and oxidants[52,53], other thermoTRPs such as TRPV1, TRPV2, and TRPM2 are redox-sensitive with the ability to discriminate between various oxidants[26,29,72–74]. Thus, post-translational modifications including channel redox state are important to consider when the intrinsic functional and structural thermosensitive properties of TRP channels are studied to understand their role as true thermoTRPs in normal physiology and pathophysiology.

Based on the analysis of single-channel open probability and intrinsic tryptophan fluorescence signaling of purified hTRPA1s, it is proposed that the CTD contains a bidirectional temperature switch priming hTRPA1 for either cold or heat, and that cold and heat responses are mediated by allosteric interactions between CTD and the S5–S6 pore region or CTD and the VSLD, respectively. Furthermore, in whole-cell patch clamp studies, the replacement of CTD-located cysteines 1021 and 1025 with alanine changed the hTRPA1 cold responsiveness, which is in line with a CTD key regulatory role of hTRPA1 cold sensitivity.

## Methods

### Recombinant protein expression and purification

The expression and purification of Δ1–688 hTRPA1 in *Pichia pastoris* (X-33, Invitrogen) was performed as described previously[7]. The gene sequence of hTRPA1 corresponding to Δ1–854 was optimized for expression in insect cells (GenScript Biotech, Piscataway, NJ, USA). The optimized gene fragment was cloned into the pTriEx-3 baculovirus donor plasmid. The amplification of the virus and expression of Δ1–854 hTRPA1 in Hi-5 and Sf9 cells (Invitrogen) were performed at the Protein Production Platform (LP3, Lund University, Sweden). The Hi-5 cell pellets were lysed using Dounce homogenizer in hypotonic buffer (10 mM HEPES, 36.5 mM sucrose, pH 7.4) supplemented with 0.1% Triton X-100 and a protease inhibitor cocktail. The Sf9 cell pellets were resuspended in membrane storage buffer (20 mM HEPES, 10% glycerol, 150 mM NaCl, pH 7.8) supplemented with a protease inhibitor cocktail and passed through an ice-cold French Pressure Cell at 18,000 psi. Unbroken cells and cell debris were removed by short low-speed centrifugation of 1500×g for 5 min at 4 °C. Membranes were collected by ultracentrifugation in a 50.2 Ti rotor (Beckman Instruments) at 45,000 rpm for 1 h at 4 °C. The resulting membrane pellet was solubilized in a membrane storage buffer supplemented with 2% Fos-Choline-14. Detergent insoluble membranes were removed by ultracentrifugation 40,000 rpm at 4 °C for 1 h. The supernatant from Hi5 cells was incubated with pre-equilibrated Ni-NTA agarose resin for 2 h, at a cold temperature. The column was washed with 10 column volumes of washing buffer (20 mM HEPES, 10% glycerol, 300 mM NaCl, pH 7.8, 0.014% Fos-Choline-14, and 50 mM imidazole). The proteins were eluted with the same buffer containing 300 mM instead of 50 mM imidazole. HisTrap HP 1 ml column (GE Healthcare) was used to purify Δ1–854 hTRPA1 proteins from solubilized Sf9 cell membranes. The column was washed with 20 column volumes of washing buffer containing 20 mM imidazole. The proteins were eluted with the same buffer containing 500 mM imidazole. These fractions were concentrated and further purified with size exclusion chromatography on a HiLoad 16/600 Superdex 200 pg column (GE Healthcare) or Sephadex G-100 connected to an ÄKTA Pure chromatography system (GE Healthcare). The fractions were collected in 20 mM HEPES, 10% glycerol, 150 mM NaCl, pH 7.8, 0.014% Fos-Choline-14. The protein

concentration was determined by measuring absorbance at 280 nm using NanoDrop spectrophotometer. The purity and expression of recombinant proteins were analysed by SDS/PAGE followed by Coomassie Blue R-250 staining and Western blotting using mouse Anti-6x-His Tag antibody (dilution 1:20,000; Becton Dickinson Pharmingen) and mouse monoclonal Anti-TRPA1 antibody (dilution 1:1000; Sigma Aldrich/MERCK). Uncropped and unprocessed scans of Western blots and SDS/PAGE gels are included in the Source Data file.

## Circular dichroism spectroscopy

Circular dichroism (CD) spectra were recorded with a Chirascan™ (Applied Photophysics). CD spectra were collected between 190 and 280 nm at 20 °C, a bandwidth of 1 nm, a collection time of 0.5 s/point, and a step size of 0.5 nm, in a 0.1 cm path-length quartz cuvette. Δ1–854 hTRPA1 at a concentration of 0.123 mg ml$^{-1}$ in 1.25 mM HEPES, 9.3 mM NaCl, 0.55% (v/v) glycerol and 0.014% Fos-Choline-14, pH 7.8, was used in CD experiments. The same buffer was used to correct the spectral background. CD measurements of Δ1–854 hTRPA1 were performed on four independent occasions and each mean spectra was obtained from 25 replicate scans. The secondary structural content was estimated using DichroWeb software (http://dichroweb.cryst.bbk.ac.uk/html/home.shtml) using the CDSSTR algorithm with the reference set SMP180t, suitable for membrane proteins[75]

## Planar lipid bilayer patch-clamp electrophysiology

These experiments were performed as originally described[7] and as follows. Detergent purified hTRPA1 were reconstituted either into preformed planar bilayers or giant unilamellar vesicles (GUVs), composed of 1,2-diphytanoyl-sn-glycero-3-phosphocholine (Avanti Polar Lipids) and cholesterol (Sigma-Aldrich) in a 9:1 ratio and produced by using the Vesicle Prep Pro Station (Nanion Technologies, Germany). Planar lipid bilayers were formed by pipetting 5 μl of either empty GUVs or protein reconstituted into GUVs on patch-clamp chips (1–2 μm, 3.5–5 MΩ resistance) which were mounted on a recording chamber. Following giga Ohm seal formation, single-channel activity was recorded using the Port-a-Patch (Nanion Technologies, Germany) at a holding potential ($V_h$) of +60 mV in a symmetrical K$^+$ solution adjusted to pH 7.2 with KOH and containing (in mM): 50 KCl, 10 NaCl, 60 KF, 20 EGTA, and 10 HEPES. The patch-clamp experiments were performed at various temperatures, for which the Port-a-Patch was equipped with an external perfusion system (Nanion Technologies) and an SC-20 dual in-line solution cooler/heater connected to a temperature-controlled (CL-100) liquid cooling system (Warner Instruments). Signals were acquired with an EPC 10 amplifier (HEKA) and the data acquisition software Patchmaster (v2x 65, HEKA) at a sampling rate of 50 kHz. The recorded data were digitally filtered at 3 kHz. Electrophysiological data were analyzed using pCLAMP/Clampfit 9 (Molecular Devices) and Igor Pro (Wave Metrics software). Data were filtered at 1000 and 500 Hz low-pass Gaussian filter for analysis and traces, respectively. The single-channel mean open probability ($P_o$) was calculated from time constant values, which were obtained from exponential standard fits of dwell-time histograms. The time constant ($\tau$) for open and closed times at various temperatures was determined by Lorentzian distribution function fit analysis of dwell-time histogram data.

The $Q_{10}$ for $P_o$ was obtained from the following equation[76,77]:

$$Q_{10} = \left(\frac{P_{o2}}{P_{o1}}\right)^{10/(T_2 - T_1)} \tag{1}$$

## Whole-cell patch-clamp electrophysiology

**Cell culture, constructs and transfection.** Human embryonic kidney 293T (HEK293T; ATCC, Manassas, VA, USA) cells were cultured in Opti-MEM I media (Invitrogen, Carlsbad, CA, USA) supplemented with 5% fetal bovine serum. The magnet-assisted transfection (IBA GmbH, Gottingen, Germany) technique was used to transiently co-transfect the cells in a 15.6 mm well on a 24-well plate coated with poly-L-lysine and collagen (Sigma-Aldrich, Prague, Czech Republic) with 200 ng of GFP plasmid (TaKaRa, Shiga, Japan) and with 300 ng of cDNA plasmid encoding wild-type or mutant human TRPA1 (pCMV6-XL4 vector, OriGene Technologies, Rockville, MD, USA). The cells were used 24–48 h after transfection. At least three independent transfections were used for each experimental group. The wild-type channel was regularly tested in the same batch as the mutants.

**Whole-cell electrophysiology and temperature stimulation.** Whole-cell membrane currents were filtered at 2 kHz using the low-pass Bessel filter of the Axopatch 200B amplifier and digitized (10 kHz) using a Digidata 1440 unit and pCLAMP 10 software (Molecular Devices, San Jose, CA, USA). Patch electrodes were pulled from borosilicate glass and heat-polished to a final resistance between 3 and 5 MΩ. Series resistance was compensated by at least 60%. Extracellular solution was Ca$^{2+}$-free and contained: 140 mM NaCl, 5 mM KCl, 2 mM MgCl$_2$, 5 mM EGTA (ethylene glycol-bis (β-aminoethyl ether)-N,N,N′,N′-tetraacetic acid), 10 mM 4-(2-Hydroxyethyl)piperazine-1-ethanesulfonic acid (HEPES), 10 mM glucose, pH 7.4 was adjusted by tetramethylammonium hydroxide. Intracellular solution contained 140 mM KCl, 5 mM EGTA, 2 mM MgCl$_2$, 10 mM HEPES, adjusted with KOH to pH 7.4. The I–V relationships were recorded by using 100-ms steps ranging from −160 to +200 mV, increment +20 mV, and a holding potential of 0 mV (shown in the inset of Fig. 9a). Only one recording was performed on any one coverslip of cells to ensure that recordings were made from cells not previously exposed to cold. For experiments with carvacrol, the voltage step protocol (shown in the inset of Fig. 9g) consisted of a 5 s steps to +80 mV from a holding potential of −80 mV, applied first in extracellular solution at 5 °C, and then in the presence of carvacrol at 25 °C. Voltage steps were applied after the channels reached equilibrium at a given temperature. A system for fast cooling and heating of solutions superfusing isolated cells under patch-clamp conditions was used as described previously[78].

## Intrinsic tryptophan fluorescence assay

Conformational changes in Δ1–688 hTRPA1 and Δ1–854 hTRPA1 were recorded on FP-8200 spectrofluorometer (Jasco, Germany) with thermostat unit, as also previously described for hTRPA1[11]. Protein was incubated in phosphate-buffered saline containing 0.014% Fos-Choline-14 at each desired temperature for 15 min and emission spectra were recorded using an excitation wavelength of 280 nm. Cold and heat experiments were done in separate experiments and for better presentation, the spectra are always normalized to the starting spectrum at room temperature. To keep the same oxygen levels in all samples and to prevent spontaneous protein aggregation/precipitation, the measurements were performed in sealed anaerobic cuvettes with constant stirring. The pH of phosphate-buffered saline was within 7.30–7.34 for the various temperatures (4–40 °C) at which the tryptophan signaling was recorded.

## Statistics

SigmaPlot 10 (Systat Software Inc., San Jose, USA), GraphPad Prism 9.1. (GraphPad Software, La Jolla, CA, USA), Igor Pro (Wave Metrics software) and CorelDraw X7 (Corel Corporation, Ottawa, Canada) were used for statistical analysis and drawing of graphs. The level of statistical significance was set at $P < 0.05$. One-way ANOVA and Dunnett's post hoc comparison tests were used for the analysis of statistical significance. Data are presented as the mean ± SEM; $n$ indicates the number of separate/independent experiments examined. The electrophysiological data were analyzed using pCLAMP 9 and 10 (Molecular Devices). Conductance–voltage (G/V) relationships were obtained from steady-state whole-cell currents measured at the end of

voltage steps. Voltage-dependent gating parameters were estimated by fitting the conductance $G = I/(V - V_{rev})$ as a function of the test potential $V$ to the Boltzmann equation:

$$G = [(G_{max} - G_{min})/(1 + \exp(-zF(V - V_{50})/RT))] + G_{min}, \quad (2)$$

where $z$ is the apparent number of gating charges, $V_{50}$ is the half-activation voltage, $G_{min}$ and $G_{max}$ are the minimum and maximum whole-cell conductance, $V_{rev}$ is the reversal potential, and $F$, $R$, and $T$ have their usual thermodynamic meanings. The peak inward conductances were related to $G_{max}$ obtained from the $G/V$ relationships.

### Reporting summary

Further information on research design is available in the Nature Research Reporting Summary linked to this article.

## Data availability

All electrophysiological patch-clamp data, tryptophan fluorescence data, and circular dichroism data supporting the conclusions drawn in this study are available within this article including Supplementary Information and as a Source Data file. Additional information and raw data of these recordings are available from the corresponding authors upon reasonable request. Plasmids of the hTRPA1 wild-type and mutants thereof investigated in this study are available from the corresponding authors upon request. The cryo-EM structures referred to in this study are available in the Protein Data Bank under the accession codes PDB 6V9W, PDB 6V9X, PDB 6PQP; and PDB 6V9Y. Source data are provided with this paper.

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

## Acknowledgements

This study was supported by the Swedish Research Council (2014–3801; P.M.Z.), the Medical Faculty of Lund University—ALF (Dnr. ALFSKANE-451751; P.M.Z.), Hjärnfonden/the Swedish Brain Foundation (FO2020-0188; P.M.Z.), Stiftelsen Olle Engkvist Byggmästare (189–290; P.M.Z.), Albert Påhlssons stiftelse (P.M.Z.), Alfred Österlunds stiftelse (P.M.Z.), the Czech Science Foundation, (grant number 22–13750S; V.V.). This

work was facilitated by the Lund Protein Production Platform at Lund University, Sweden (http://www.lu.se/lp3). This project has received funding from European Union's Horizon 2020 research and innovation program under grant agreement No. 101004806 (ProLinC, a node in the MOSBRI biophysics infrastructure).

## Author contributions
L.M. and P.M.Z. conceptualized and directed research; L.M., P.M.Z., M.R.F., and V.V. designed research; L.M., V.S., V.K.M., T.V., and M.R.F. performed research; L.M., V.S., V.K.M., M.K., T.V., M.R.F., V.V., and P.M.Z. analyzed data; P.M.Z. drafted the paper and L.M., V.V., and P.M.Z. wrote the paper.

## Funding

## Competing interests
The authors declare no competing interests.
