## [Peer Review File · Nature Communications]

REVIEWER COMMENTS

Reviewer #1 (Remarks to the Author):

In this manuscript, Moparthy et al. study the mechanisms underlying temperature-dependent gating of the cation channel TRPA1. In a first part, they use truncated versions of the channel reconstituted into lipid bilayers or studied in isolation in a tryptophan fluorescence assay, whereas in the second part, they study wild type channels and specific point mutations after expression in HEK293 cells.

Based on their cumulative results, the authors propose a model where the C-terminal domain contains cold and heat sensitive domains that are allosterically coupled to the pore. In their model, the voltage-sensitive domain (S1-S4) has an inhibitory effect on cold-induced gating, but is required for signal transduction from the heat sensitive domain to the pore.

Despite a significant body of work (and even a recent Nobel prize) the molecular/structural basis of temperature-dependent gating of (TRP) channels is still poorly understood, and therefore studies like this one can have their value. However, at this point, the provided data are not fully convincing, and it is unclear whether the results are of any relevance for TRPA1 in its native conformation.

Main points:

- 1) In their bilayer and tryptophan fluorescence experiments, the authors make use of isolated forms of hTRPA1 lacking the main part of the N terminus ($\Delta 1-688$; this construct was already used in Mparthy et al. PNAS, 2014) or even the entire N terminus + the first four transmembrane domains ($\Delta 1-854$). Considering the detailed structural information that has been acquired for TRPA1 (see e.g. ref 47), showing tight interactions between N-terminal (e.g. the Coupling Domain), C-terminal (IFH and TRPL domain) and the S4-S5 linker, it is surprising (not to say, hard to imagine) that a construct that lacks S4 and everything N terminal of it could be functional. The authors should explain/discuss how, in the absence of so many intersubunit interactions, this construct could still assemble into a functional channel. There is also no real (biochemical) evidence to show that this construct actually forms a tetramer. Do these constructs yield whole-cell currents when expressed in HEK293 cells?
- 2) Even if one accepts that the N-terminally truncated construct lacking the first four transmembrane regions forms an actual tetrameric channel, it is unclear how it can teach us anything about the temperature-dependent gating of TRPA1. There is a huge and poorly discussed disconnect between the dazzling Q10 value of 1000 for cold activation of $\Delta 1-854$ in lipid bilayer and the very modest 2-fold increase upon cooling wild type TRPA1 (i.e. a Q10 of 2?) in the whole-cell recordings at -80 mV (and actual a decrease in current at positive voltages). Therefore, the question arises whether these processes have anything to do with each other?
- 3) The single-channel data are not really analysed in detail (only Popen is provided, along with single channel conductances in the Tables). When looking at the individual traces, it seems that there are many other parameters that are different at the different temperatures, and which could be investigated using a more detailed analysis of open and closed times, channel bursts etc. For instance, the single channel behaviour of $\Delta 1-854$ (Fig. 2A) seems to change dramatically between 30 and 35 degrees, from relatively long open and closed times ($\sim 100-1000$ ms at 30 degrees) to a constant flickering between open and closed states with unresolved but much shorter open and closed times at 35 degrees. So the data for Popen do not reflect this quite dramatic effect on gating. Just below, the trace at 40 degrees does not seem to be a steady-state condition – instead, the first second of the trace shows an open probability of ~ 0 , and then the open probability seems to gradually increase in the following ~ 10 s to reach a value of ~ 1 at the end of this trace.
- 4) Also, the noise level seems to differ quite substantially and non-monotonously between

the different traces (e.g. compare the traces at 17 and 30 degrees with those at 20 and 25 degrees for the shortest construct). It seems that the traces are from selected, individual experiments rather than traces from one individual experiment/membrane patch at different temperatures (this would be ideal)? It would be much more convincing to show the response of one channel to a temperature ramp.

5) A few other aspects of the planar lipid experiments look a bit unnatural:

*In Figure 1C, the effect of the pore blocker RR seems to be a gradual decrease in channel conductance developing over seconds, rather than the expected abrupt closing. Note also how in this trace, the open probability is ~ 0.05 for the first ~ 30 s, and then abruptly shifts to an open probability of ~ 0.95 .

*In Figure 2C, HC030031 causes a quite substantial drop of the zero current level.

* Single channel conductances for the truncated channels (Tables 1 and 2) are also quite peculiar: For $\Delta 1-688$, conductance changes non-monotonously between 15 pS at 10 degrees, 68 pS at 20 degrees, 27 pS at 35 degrees and 54 pS at 40 degrees. What could be the reason for such variations in conductance? How do these conductances relate to what is seen for TRPA1 in a cellular environment?

6) The whole-cell patch-clamp experiments in Figure 6 are not ideal to analyse the thermal sensitivity and voltage dependence of the cysteine mutants. To study voltage dependence, it is necessary to use voltage steps instead of 200-ms ramps as used here, since at lower temperatures it takes >200 ms to reach voltage-dependent steady-state relaxation (see e.g. Karashima et al. PNAS, 2009; Sinica et al. Cells, 2019). Therefore, the observed differences in outward current and/or rectification may reflect slower current kinetics, irrespective of altered thermal or voltage sensitivity. In order to allow proper interpretation of these data, a more detailed analysis as in Sinica et al. (2019) is warranted. And, as mentioned above, it is necessary to provide evidence and discussion to link the very modest cold sensitivity observed here with the extreme cold sensitivity observed in the bilayer experiments.

Other points:

1) Using the authors' formula (page 11), Q_{10} values for cold activation cannot be >1 . For instance, for the change in open probability for $\Delta 1-854$ between 20 and 15 degrees, the formula yields: $Q_{10} = (0.92/0.03)^{(10/(15-20))} = 0.001$.

2) Page 6 and 8 - "break": probably meant is "brake"?

Reviewer #2 (Remarks to the Author):

In the manuscript the authors aim to unravel the contribution of different domains to cold and heat thermo-sensitivity of human TRPA1. To this end, two truncated variants were constructed: removal of ankyrin repeats and pre-S1 domain, D1-688 (which was studied before already), and additionally removal of the voltage-sensor like domain, VSLD, in D1-854). These were purified and their function studied via planar bilayer channel recordings. Additionally, structural changes were followed by tryptophan fluorescence. The role of cysteines was studied in whole cell patch clamp measurement for the wt. From these experiments the authors conclude, that "the CTD contains a bidirectional temperature switch priming hTRPA1 for either cold or heat, and that cold and heat responses are mediated by allosteric interactions between CTD and the S5-S6 pore region or CTD and the VSLD, respectively."

The conclusion is based on the following observations:

a) Deletion of the N-terminal domain up to AS 688 did not abolish cold and heat-sensitivity, but increased the steepness of response in both directions (increased q_{10} -values for cold and heat, respectively). Further deletion of S1-S4 even further increased the sensitivity to cold, but reduces sensitivity to heat.

=> Indeed, from this it can be deduced, that the presence of the ankyrin-part down-

regulates the response to both heat and cold, while presence of VSLD down-regulates cold, but up-regulates heat response.

b) In an attempt to identify structural changes, the response of the tryptophans to cold and heat was compared for both constructs. This approach is hampered by a number of problems:

i) fluorescence intensity is intrinsically temperature dependent. However, the extent of thermal quenching depends on the local environment. Therefore, if constructs with different number of trps are compared, the result might be different, just due to different environments of the single trps. Thus, to disentangle this from the effect of conformational changes is nearly impossible.

A way out of this might be the strategy followed in some denaturation experiments. Here, in order to grab the slight shift in the emission wavelength (which also seems to be present here) a kind of mean wavelength is calculated (see Royer et al, Protein Science 2.11 (1993): 1844-1852.). This might well work here, with the exception of the measurement at high temperatures in case of D1-688, where the spectrum is dominated by non-trp contributions.

ii) the purification level of the two variants seems to be quite different (compare the SDS-PAGE in this MS with ref. 7). This further adds to problem i)

iii) The drop observed upon heat exposure seems to be extremely strong in case of D1-688. Did the authors check for reversibility, loss of protein due to sticking to Eppendorf-tubes etc?

=> Thus, the information content of this part is possibly rather limited, and one might wonder, whether this warrants the extensive discussion addressing this point. The experiment, which most clearly shows a conformational transition is the cold-response of D1-854, showing a sudden drop at around 20°C. Also the cold-response of D1-688 probably reflects some structural changes, since intrinsic thermal response of trp would actually lead to an increase in intensity. However, what has to be also kept in mind, that temperature changes shift the pH, and different constructs might respond differentially to this effect.

c) TCEP inhibits cold responses, implying the formation of cys-bridges could be important. Thus, the role of three cysteines were studied in whole-cell patch clamp at 15 and 25°C. Each mutation leads to some effect, but each is different.

=> The main conclusion from this is, that these observations support an allosteric model.

Thus overall, the findings are interesting, but possibly in case of the trp-fluorescence measurement not as robust as the authors assume. The problems described above should be addressed and mentioned. The discussion should be shortened and straightened. In the present style, it is difficult to follow.

Specific comments

1) Tryptophan fluorescence studies:

a) The structural changes are followed based on tryptophan fluorescence of detergent solubilized protein. Often the structure of membrane proteins differ in detergent micelles compared to lipid membranes. Are there any studies showing that this approach is valid, and can be extrapolated to the lipid environment?

b) Furthermore, due to the rather strong temperature dependence of pH in case of Hepes ($dpH/dT = -0.022$) also could add to some of the effects observed. If the pH was set to pH 7.8 at 25°C, the pH will be around 8.2 at 7°C and below 7.5 at 40°C. This should at least be mentioned.

c) Why is there an emission band around 420 nm in the spectra of heated Δ 1-688? In the

cooled spectra this band is missing. Actually, also the shape at the low wavelength side and the overall intensity is different. Are these two different preparations with different levels of purity?

d) The hump at 305 nm could be water-Raman scattering peak or tyrosine. In case of the former, it should be visible also in other spectra with similar intensities. Are all spectra shown in fig.3 measured at exactly the same settings of the fluorimeter, so that the intensities can be compared? If so, then the hump is probably stemming from tyrosines.

2) Patch-Clamp:

The $\Delta 1-688$ mutation leads to higher current density (in magnitude) at negative holding potential, and lower at positive voltage. What about $\Delta 1-854$?

Minor comments

1) When comparing the results of $\Delta 1-688$ and $\Delta 1-854$, the differences are a bit difficult to grab as described in the text. It might help to start from the minimum of activity in each case (22°C and 20°C, respectively) and describe the change in activity by heating or cooling. This is done by reporting Q10 values, but it is not always clear, what the reference temperature is (e.g. legend in Fig. 1, last sentence: Q10 cold 0 56, Heat (20 to 30°C) = 29. Is q10 cold refering to 10 to 20°C?

2) Why is it stated (results, first paragraph) that $\Delta 1-854$ has lost its heat sensitivity? There is an increase in channel activity between 20 and 25°C. Thus, one observes a strongly decreased heat sensitivity.

3) Figure 4: please add, that trps are in blue, and cysteine in yellow. The letters of the amino acids are pretty small and hard to decipher.

4) Figure 3: It would be easier for the reader to follow, if the sub-figures were numbered A,B,C etc.

5) Why is the temperature range color coded for one mutant, but not for the other? (blue/red areas in fig. 2 and 3)

6) Page 5, second paragraph: "Repeated voltage ramps....in control extracellular solution.." Meaning of control solution unclear.

7) Page 5, second paragraph, last third: "C1021A and C1025A were reduced by 0.5 +/- 0.3 ..."

Typo in the error?

8) Page 5, first sentence in discussion: "been shown to response directly"...please clarify "directly". Is this meant in contrast to others, where the thermo-sensing is performed by other proteins, which interact with TRP channels?

9) page 7, first paragraph, last third: "nadir" : a typo, I guess.

10) Fig. 7: CTD is referred to as transition state. A transition state is clearly defined (see enzymatics); an alternative would be "intermediate state", maybe.

11) Page 6: second paragraph, middle, discussion of the effect of removing the VSLD, loss of heat sensitivity: "this latter indicating heat sensitive structures within the VSLD or its uncoupling from a heat-sensitive CTD". The first part is clear; but the second part not really. I assume, that the authors want to say, that with intact VSLD, it is coupled to the heat-sensitive CTD, supporting the response to heat? Please clarify.

12) Of note, the single channel conductance of the $\Delta 1-854$ mutant is much less temperature dependent, while the $\Delta 1-688$ has a strong, non-monotonous dependence. This is not mentioned at all.

Reviewer #3 (Remarks to the Author):

In this manuscript, Moparathi et al. studied the role of different structural domains in the temperature sensitivity of TRPA1. The authors claim that the heat sensitivity of TRPA1 is dependent on the VSLD and CTD and the cold sensitivity is dependent on the CTD and pore domain. In general, these findings are quite interesting and new. However, from my point

of view, several flaws hinder this manuscript from being suitable for publication. Below are a few of the most critical issues:

1) The authors only use truncated TRPA1 protein for their lipid bilayer studies. Although the difficulty in purifying the full TRPA1 protein is understood, the authors' conclusion must be supported by demonstrating that in their hands, using the same recording methodologies, the full length and the truncated share the relevant features. Moreover, considering that previous studies showed that TRPA1 temperature sensitivity requires the N-ARD (the truncated part) (PMID: 21930928), the recording of the full length in lipid bilayer becomes critical for this manuscript.

2) The authors show that the truncated TRPA1 shifted its conductance between 15 and 40 degrees (both at +60mV) by almost a factor of 2. How comes? Do the authors suggest that the pore is only partially open in the cold? These results require explanation and normalization if the authors suggest that its ion mobility issues.

3) It is not clear why several points in the tables are from previous studies. The authors should repeat the experiments or just cite the previous study. The current presentation is misleading. The reader might think that all the data was acquired in the same experimental session.

4) The bars in figure 6E are misleading and do not reflect the presented data in the figure. It is very tough to review this figure due to this problem.

5) The single-channel recordings in figure 2C look like multiple channels. Also, the authors should explain and show the Q10 calculation. It seems that Q10 of 1000 should be explained in detail.

REVIEWER COMMENTS

Reviewer #1 (Remarks to the Author):

In this manuscript, Moparthi et al. study the mechanisms underlying temperature-dependent gating of the cation channel TRPA1. In a first part, they use truncated versions of the channel reconstituted into lipid bilayers or studied in isolation in a tryptophan fluorescence assay, whereas in the second part, they study wild type channels and specific point mutations after expression in HEK293 cells. Based on their cumulative results, the authors propose a model where the C-terminal domain contains cold and heat sensitive domains that are allosterically coupled to the pore. In their model, the voltage-sensitive domain (S1-S4) has an inhibitory effect on cold-induced gating, but is required for signal transduction from the heat sensitive domain to the pore. Despite a significant body of work (and even a recent Nobel prize) the molecular/structural basis of temperature-dependent gating of (TRP) channels is still poorly understood, and therefore studies like this one can have their value. However, at this point, the provided data are not fully convincing, and it is unclear whether the results are of any relevance for TRPA1 in its native conformation.

Authors: We are thankful to the reviewer for the very valuable and constructive criticism on our study. We are happy to know that the reviewer finds the study of interest, and hope that we have satisfactorily addressed below all points raised by the reviewer.

Main points:

1) In their bilayer and tryptophan fluorescence experiments, the authors make use of isolated forms of hTRPA1 lacking the main part of the N terminus ($\Delta 1-688$; this construct was already used in Mparthi et al. PNAS, 2014) or even the entire N terminus + the first four transmembrane domains ($\Delta 1-854$). Considering the detailed structural information that has been acquired for TRPA1 (see e.g. ref 47), showing tight interactions between N-terminal (e.g. the Coupling Domain), C-terminal (IFH and TRPL domain) and the S4-S5 linker, it is surprising (not to say, hard to imagine) that a construct that lacks S4 and everything N terminal of it could be functional. The authors should explain/discuss how, in the absence of so many intersubunit interactions, this construct could still assemble into a functional channel. There is also no real (biochemical) evidence to show that this construct actually forms a tetramer. Do these constructs yield whole-cell currents when expressed in HEK293 cells?

Authors: We believe that TRPA1 is equipped over time with important accessories such as the N-ARD. We believe that $\Delta 1-854$ hTRPA1 has preserved basic gating mechanisms as also shown for NaVs (Shaya et al PNAS 2011). We certainly do not exclude the importance of N-ARD and VSLD in modulating TRPA1

function. No doubt, cryo-EM has added new insights into TRP channel structure-function analysis. However, functional experiments will continue to guide and validate cryo-EM studies. We expect an equilibrium between different oligomeric states including tetramers of purified TRPA1 in detergent. However, in the bilayer electrophysiology experiments, it is difficult to imagine that we are studying currents with TRPA1 oligomeric states less than tetramers. Furthermore, the shared pharmacology of hTRPA1, $\Delta 1-688$ hTRPA1 and $\Delta 1-864$ hTRPA1 with regard to TRPA1 inhibition of chemo- mechano- and thermosensory responses (Moparathi et al *PNAS* 2014, *Sci Rep* 2016; *Cell Calcium* 2020; Moparathi & Zygmunt *Cell Calcium* 2020), especially by the **TRPA1 antagonist HC030031** (McNamara et al *PNAS* 2007; Gupta et al *Sci Rep* 2016), add key evidence of $\Delta 1-854$ hTRPA1 being a functional tetrameric ion channel. A proper folding and tetrameric formation of $\Delta 1-854$ hTRPA1 is further supported by CD-spectra and size exclusion chromatography analyses included in the revised manuscript (now Fig. 1). Exactly how $\Delta 1-854$ hTRPA1 assemble can, however, only be speculated on but as generally assumed, subunits should be held together by mainly hydrogen bonds and van der Waals forces. Finally, our protein dissection strategy is in line with the study by Shaya et al (*PNAS* 2011) in which the structure-activity relationship of NaVs, incorporated into artificial lipid bilayers for electrophysiology recordings, was analyzed using truncated proteins only containing S5-S6.

Regarding heterologous expression of truncated hTRPA1s, it is unfortunate that we (Berrouit et al *Nat Commun* 2017) and others (Nilius et al *J Physiol* 2011) have found that N-ARD truncated TRPA1 cannot be expressed/inserted into the cell membrane, and thus our approach of using purified truncated hTRPA1s offers unique possibilities to study polymodal gating of TRPA1 independent of the N-ARD as well as the S1-S4 voltage sensing-like domain. However, our aim to address the role of the hTRPA1 intracellular C-terminus in temperature, mechanical and calcium responses (Moparathi & Zygmunt *Cell Calcium* 2020; Moparathi et al *Cell Calcium* 2020), has so far been hampered because of failed expression and purification of C-terminal truncated TRPA1s. This may also be an issue in transfected cells.

2) Even if one accepts that the N-terminally truncated construct lacking the first four transmembrane regions forms an actual tetrameric channel, it is unclear how it can teach us anything about the temperature-dependent gating of TRPA1. There is a huge and poorly discussed disconnect between the dazzling Q10 value of 1000 for cold activation of $\Delta 1-854$ in lipid bilayer and the very modest 2-fold increase upon cooling wild type TRPA1 (i.e. a Q10 of 2?) in the whole-cell recordings at -80 mV (and actual a decrease in current at positive voltages). Therefore, the question arises whether these processes have anything to do with each other?

Authors: We trust our findings will stimulate the search for possible specialized channel thermosensor modules and how they are linked to channel pore gating of TRPA1 and perhaps other TRP channels. In the case of hTRPA1 we can clearly conclude that cold and heat sensitivity involve separate channel structures and gating mechanisms. This may also be true in intact cellular systems where redox and other posttranslational modifications could promote cold and heat sensitive conformations as we have demonstrated for purified hTRPA1 in the present study and previously (Moparathi et al *Sci Rep* 2016). A similar conclusion regarding TRPA1 heat sensitivity has also been reached by Vandewauw et al (*Nature* 2018). The Q10 values calculated for the purified hTRPA1s are primarily relevant for comparison between the various purified hTRPA1s, illustrating the relationship between hTRPA1 structure and its intrinsic ability to respond to changes in temperature. Due to failure of heterologous cellular expression of N-ARD truncated hTRPA1s we cannot explore whether such Q10 values reported here can be obtained in intact cells. However, similar Q10 values may not appear in cells as there are in addition to environmental factors other temperature-dependent mechanisms (e.g., potassium channels) that may counteract TRPA1 activity. Only a few studies have determined Q10 values for mammalian TRPA1 cold responses below 17°C in intact cells, providing Q10 values of ~0.1 (Story et al *Cell* 2003; Sawada et al *Brain Res* 2007; Karashima et al *PNAS* 2009). Clearly, as pointed out by the

reviewer the Q10 value of hTRPA1 expressed in HEK293 cells in the present study is also far from that of purified hTRPA1s in bilayer experiments. It seems as if TRPA1 thermal quantitative properties are not easily transferred between various experimental models. Likewise, Q10 values for TRPV1, TRPM8 and TRPM3 as purified channels are not always in good agreement with Q10 values obtained for these channels in a cellular environment. Surprisingly, TRPM3 lacks the strong, if any, intrinsic temperature sensitivity as reported in cells, when inserted into bilayers and could easily have been disqualified as a thermoTRP (Vriens et al *Neuron* 2011; Uchida et al *Faseb J* 2016). This could be due to lack of appropriate lipid environment, as the presence of phospholipids influence the TRPV1 and TRPM8 in bilayer recordings (Cao et al *Nature* 2013; Zakharian et al *JNS* 2010; Sun & Zakharian *JBC* 2015). Indeed, Q10 differ substantially for both TRPV1 and TRPM8 compared to whole-cell recordings as discussed (Zakharian et al *JNS* 2010; Sun & Zakharian *JBC* 2015). In this context it is interesting that TRPV1 could not alone explain the well-defined noxious $T_{\text{threshold}}$ of 43°C in rodents (Caterina et al *Science* 2000; Davis et al *Nature* 2000). Indeed, TRPV1 acts in concert with TRPA1 and TRPM3 to set this noxious heat temperature threshold (Vandewauw *Nature* 2018). Thus, there is not always a perfect translational match when studying thermoTRPs. If the aim is to gain knowledge of TRPA1 intrinsic temperature-structure gating properties at a molecular level, then studies of purified TRPA1s as in the present study are key. We have added a paragraph in the discussion commenting on the discrepancy in Q10 values between purified hTRPA1s and whole-cell recordings.

3) The single-channel data are not really analysed in detail (only Popen is provided, along with single channel conductances in the Tables). When looking at the individual traces, it seems that there are many other parameters that are different at the different temperatures, and which could be investigated using a more detailed analysis of open and closed times, channel bursts etc. For instance, the single channel behaviour of $\Delta 1-854$ (Fig. 2A) seems to change dramatically between 30 and 35 degrees, from relatively long open and closed times (~100-1000 ms at 30 degrees) to a constant flickering between open and closed states with unresolved but much shorter open and closed times at 35 degrees. So the data for Popen do not reflect this quite dramatic effect on gating. Just below, the trace at 40 degrees does not seem to be a steady-state condition – instead, the first second of the trace shows an open probability of ~0, and then the open probability seems to gradually increase in the following ~10 s to reach a value of ~1 at the end of this trace.

Authors: Our aim was to study the relationship between TRPA1 structure and temperature responsiveness, and as such it is highly adequate to primarily analyze the single-channel open probability, also allowing comparison with our previous studies on hTRPA1, $\Delta 1-688$ hTRPA1 and AgTRPA1 temperature sensitivity (Moparthi et al *PNAS* 2014; Moparthi et al *Sci Rep* 2016; Survery et al *JBC* 2016). We agree with the reviewer that our recordings may contain additional important information on temperature gating. Thus, we have included analysis of $\Delta 1-854$ hTRPA1 open and closed time (Supplementary Fig. 1). We hope to be able to perform similar analysis of channel kinetics as well as channel bursts etc., for all three constructs in the future, if resources allow. The traces and their corresponding histograms are part of longer recordings used for determining single-channel open probability and conductance, and chosen to fairly illustrate in a single short trace that channel activity is not homogenous over the entire recording time and that different patterns of activity occurred within the entire period of analysis. Thus, there is sometimes intermittent burst activity but the overall analysis of channel open probability is accurately described by the calculated P_o values presented in Tables 1 & 2 and corresponding graphs. We hope the reviewer agrees that we have demonstrated that cold and heat channel activity is indeed gated differently by separate protein structures, which is clearly demonstrated also by the additional analysis of $\Delta 1-854$ hTRPA1 open and closed time (Supplementary Fig. 1, Supplementary Table 1).

4) Also, the noise level seems to differ quite substantially and non-monotonously between the different traces (e.g. compare the traces at 17 and 30 degrees with those at 20 and 25 degrees for the

shortest construct). It seems that the traces are from selected, individual experiments rather than traces from one individual experiment/membrane patch at different temperatures (this would be ideal)? It would be much more convincing to show the response of one channel to a temperature ramp.

Authors: This is true, and could be as a result of using different batches of bilayers although the fabrication of lipid bilayers is standardized and the quality is very consistent. Technically, it is difficult to obtain the whole temperature range studied in a single “membrane patch”. For the reviewer we have included a trace from a continuous recording (3.5 min), as has previously also been shown for hTRPA1, $\Delta 1-688$ hTRPA1 and AgTRPA1 (Moparathi et al *PNAS* 2014; Moparathi et al *Sci Rep* 2016; Survery et al *JBC* 2016). As also illustrated by the trace, there are sometimes few channel openings around Po nadir although not with enough of events to calculate Po. The same is true for hTRPA1, $\Delta 1-688$ hTRPA1 and $\Delta 1-854$ hTRPA1 at 20 - 22°C as well as for AgTRPA1 and $\Delta 1-776$ AgTRPA1 at 22 - 25°C (Moparathi et al *PNAS* 2014; Moparathi et al *Sci Rep* 2016; Survery et al *JBC* 2016). Our experimental set-up does not allow rapid automated temperature ramps. However, as discussed for our whole-cell recordings (point 6 below), steady-state conditions may be advantageous also when studying temperature responses as we have done in this study.

5) A few other aspects of the planar lipid experiments look a bit unnatural: *In Figure 1C, the effect of the pore blocker RR seems to be a gradual decrease in channel conductance developing over seconds, rather than the expected abrupt closing. Note also how in this trace, the open probability is ~ 0.05 for the first ~ 30 s, and then abruptly shifts to an open probability of ~ 0.95 . *In Figure 2C, HC030031 causes a quite substantial drop of the zero current level. *Single channel conductances for the truncated channels (Tables 1 and 2) are also quite peculiar: For $\Delta 1-688$, conductance changes non-monotonously between 15 pS at 10 degrees, 68 pS at 20 degrees, 27 pS at 35 degrees and 54 pS at 40 degrees. What could be the reason for such variations in conductance? How do these conductances relate to what is seen for TRPA1 in a cellular environment?

Authors: These are interesting and intriguing observations. We have noticed that the onset of RR effect can be slower compared to HC030031 also in our previous studies. We do not have any obvious explanation for this, but maybe as pore blocker RR is more dependent than HC030031, with binding sites outside the pore (Gupta et al *Sci Rep* 2016), on the pore dynamics including size. Interestingly, pore dilation occurs in TRPA1, indicating that the TRPA1 pore structure is dynamic (Banke et al *Mol Pain* 2009). Maybe the gating of TRPA1 as reflected by less variability in Po is more robust than its pore properties as reflected by greater variability in conductance. Indeed, the conductance, but not Po, decreased when hTRPA1 was repeatedly exposed to 15°C (Moparathi et al *PNAS* 2014). It is noteworthy that TRPA1 single-channel conductance values differ greatly depending on stimuli, species, cell and test conditions including ion composition and co-factors such as calcium and polyphosphates (Zygmunt & Högestätt, *Handb Exp Pharmacol* 2014, chapter 5 including Table 1).

Although, our hTRPA1s when purified and reconstituted into bilayers may display various single-channel channel opening levels in response to the same temperature as well as the same ligand (Moparthy et al *PNAS* 2014), collectively the conductance values are within those reported for whole-cell configurations as well as isolated cell membrane patches (Zygmunt & Högestätt, *Handb Exp Pharmacol* 2014.) For example, the single-channel conductance value was 40 pS at 15°C compared to 91 pS at 25°C and a hp of +50 mV (Karashima et al *PNAS* 2009). Thus, there is no simple biophysical fingerprint of hTRPA1 with regard to single-channel conductance. In the present study, we have only analyzed the most frequent single-channel conductance state, allowing us to make comparisons with our previous studies. Importantly, this “main” single-channel conductance of Δ 1-854 hTRPA1 is overall similar to “main” hTRPA1 and Δ 1-688 hTRPA1 conductance values within the temperature interval of 25 - 40°C (hp +60 mV), providing evidence that Δ 1-854 hTRPA1 truncation did not affect the pore function but the sensitivity to heat. However, the reviewer may be right that VSLD not only affect channel activity as measured by open probability but also pore function as indicated by non-monotonous conductance. Clearly, it is needed to follow up with future studies and more detailed analysis to better understand channel behavior with regard to conductance and pore properties, especially as we have previously shown that the conductance for ligands and pressure is dependent on the N-ARD as well as voltage (Moparthy et al *PNAS* 2014; Moparthy & Zygmunt *Cell Calcium* 2020). Thus, it would not be surprising if the N-ARD intramolecularly also modulates the hTRPA1 pore function in response to temperature, which these results could indicate.

Fig. 1C (now Fig. 2c). Regarding abrupt shift of P_o , see reply to point 3 above.

Fig. 2C (now Fig. 3c). Regarding substantial drop of the zero current level at application of HC030031, this could be because of the very high frequency channel activity not allowing proper detection of baseline, unless channel activity is fully blocked. We have observed this phenomenon earlier as shown e.g., for Δ 1-688 hTRPA1 when exposed to cinnamaldehyde and subsequently HC030031 (Fig. 3B, Moparthy et al *PNAS* 2014). However, it could also be due to change of perfusion solution, which sometimes initially can affect baseline, although without influence on the analysis of P_o and conductance.

6) The whole-cell patch-clamp experiments in Figure 6 are not ideal to analyse the thermal sensitivity and voltage dependence of the cysteine mutants. To study voltage dependence, it is necessary to use voltage steps instead of 200-ms ramps as used here, since at lower temperatures it takes >200 ms to reach voltage-dependent steady-state relaxation (see e.g. Karashima et al. *PNAS*, 2009; Sinica et al. *Cells*, 2019). Therefore, the observed differences in outward current and/or rectification may reflect slower current kinetics, irrespective of altered thermal or voltage sensitivity. In order to allow proper interpretation of these data, a more detailed analysis as in Sinica et al. (2019) is warranted. And, as mentioned above, it is necessary to provide evidence and discussion to link the very modest cold sensitivity observed here with the extreme cold sensitivity observed in the bilayer experiments. Other points: 1) Using the authors' formula (page 11), Q_{10} values for cold activation cannot be >1. For instance, for the change in open probability for Δ 1-854 between 20 and 15 degrees, the formula yields: $Q_{10} = (0.92/0.03)^{(10/(15-20))} = 0.001$. 2) Page 6 and 8 - “break”: probably meant is “brake”?

Authors: We agree with the reviewer that ramps are not ideal for studying the complex effects of voltage and temperature and were used more to maintain compatibility with our previous study on hTRPA1 (Moparthy et al *Sci Rep* 2016). As suggested, we have performed additional experiments using voltage steps from -160 mV to +200 mV, applied from a holding potential of 0 mV, and the conductance-to-voltage (G/V) relationships were compared at 25 °C and 15 °C. These new data (shown in Figure 6a-f of the revised manuscript) confirm our previous findings that the C856A mutation shifts the half-maximum activation voltage (V_{50}) at 25°C rightward, which we interpret as an impaired allosteric coupling of the putative voltage sensor to the gate of the channel (Sinica & Vlachova, *Physiol*

Res 2021). Cold shifted V_{50} by about -28 mV in the wild-type but only by about -20 mV in C856A. On the other hand, mutations at C1021A and C1025A caused a more pronounced leftward shift in the G/V curve at 15°C compared with the wild-type channels (by about -32 mV, -37 mV and -46 mV in C1021A, C1025A and C1021A/C1025A), suggesting involvement of these two cysteines in voltage- and temperature-dependent gating. To further examine the extent of cold-dependent activation in the cysteine mutants, we also performed steady-state experiments using a protocol similar to that used recently for TRPM8 by Yang et al. (*PNAS* 2020, 117, 8633-8638). Thus, we measured cold-dependent activation under conditions in which the putative voltage sensor was supposed to be only partially activated (+80 mV) and the putative cold sensor strongly activated (5 °C). The responses were related to the currents produced by the presence of a saturating concentration of the non-covalent agonist of TRPA1 carvacrol (100 μM). Carvacrol-induced maximum current densities measured at +80 mV were not significantly different among all the constructs, however, C1021A and C1021A/C1025A exhibited significantly decreased relative response to cold at +80 mV (Fig. 7 g, h). In our opinion, this result further supports the involvement of the three cysteines in voltage and cold-dependent gating.

Q10 cold values are now given according to the formula. Break is brake.

Reviewer #2 (Remarks to the Author):

Authors: We are thankful to the reviewer for the very valuable and constructive criticism on our study. We are happy to know that the reviewer finds the study of interest, and hope that we have satisfactorily addressed below all points raised by the reviewer.

In the manuscript the authors aim to unravel the contribution of different domains to cold and heat thermo-sensitivity of human TRPA1. To this end, two truncated variants were constructed: removal of ankyrin repeats and pre-S1 domain, Δ1-688 (which was studied before already), and additionally removal of the voltage-sensor like domain, VSLD, in Δ1-854). These were purified and their function studied via planar bilayer channel recordings. Additionally, structural changes were followed by tryptophan fluorescence. The role of cysteines was studied in whole-cell patch-clamp measurement for the wt. From these experiments the authors conclude, that “the CTD contains a bidirectional temperature switch priming hTRPA1 for either cold or heat, and that cold and heat responses are mediated by allosteric interactions between CTD and the S5-S6 pore region or CTD and the VSLD, respectively.” The conclusion is based on the following observations:

a) Deletion of the N-terminal domain up to AS 688 did not abolish cold and heat-sensitivity, but increased the steepness of response in both directions (increased q_{10} -values for cold and heat, respectively). Further deletion of S1-S4 even further increased the sensitivity to cold, but reduces sensitivity to heat. => Indeed, from this it can be deduced, that the presence of the ankyrin-part down-regulates the response to both heat and cold, while presence of VSLD down-regulates cold, but up-regulates heat response.

b) In an attempt to identify structural changes, the response of the tryptophans to cold and heat was compared for both constructs. This approach is hampered by a number of problems:

i) fluorescence intensity is intrinsically temperature dependent. However, the extent of thermal quenching depends on the local environment. Therefore, if constructs with different number of trps are compared, the result might be different, just due to different environments of the single trps. Thus, to disentangle this from the effect of conformational changes is nearly impossible.

Authors: We agree that it may not be as straight forward when comparing Δ1-688 hTRPA1 and Δ1-854 hTRPA1 because of an unequal number of trps. Therefore, we have added the following note at

the end of that paragraph. “It should be noted that the comparison of $\Delta 1$ -688 hTRPA1 and $\Delta 1$ -854 hTRPA1 fluorescence may be complicated, since the two remaining tryptophans in $\Delta 1$ -854 hTRPA1 that are located within the CTD may experience a different local environment in the absence of VSLD. Nevertheless, our results clearly show that cold and heat responses within the same construct involve different channel conformational changes. Also, there is a substantial rearrangement of the CTD at the interface between warm and cold temperatures as shown in $\Delta 1$ -854 hTRPA1.”

A way out of this might be the strategy followed in some denaturation experiments. Here, in order to grab the slight shift in the emission wavelength (which also seems to be present here) a kind of mean wavelength is calculated (see Royer et al, Protein Science 2.11 (1993): 1844-1852.). This might well work here, with the exception of the measurement at high temperatures in case of $\Delta 1$ -688, where the spectrum is dominated by non-trp contributions.

Authors: As the reviewer points out this will not work for high temperatures which would prevent us from plotting and better understand the heat responses using such a strategy. All the potential issues that the reviewer raises could indeed happen, but there are some experiments that may prove that is not the case. Although we have to rely on identical experiments with the full length hTRPA1 sharing a similar fluorescence profile (including the hump) with $\Delta 1$ -688 hTRPA1, the cold- and heat-induced fluorescence changes were blunted (cold) and substantially inhibited (heat) by DTT (Moparthi et al *Sci Rep* 2016). This is not in line with a non-specific cold and heat denaturation of hTRPA1 and most likely also $\Delta 1$ -688 hTRPA1. Indeed, as shown below by CD spectroscopy, the shape and magnitude of the $\Delta 1$ -688 hTRPA1 signal is intact between 25 - 50 °C suggesting that there is no gross protein denaturation that can explain the almost complete quenching of the $\Delta 1$ -688 hTRPA1 trp fluorescence signal in response to heat. Furthermore, as shown below for the knowledge of the reviewer, DTT almost completely inhibited $\Delta 1$ -854 hTRPA1 cold-induced trp fluorescence changes (n = 3), but had no effect on heat-induced trp fluorescence changes up to 43 °C (n = 2). However, heat denaturation cannot explain the intact trp fluorescence changes in the presence of DTT as $\Delta 1$ -854 hTRPA1 was stable at temperatures reaching 50 °C, as measured by CD spectroscopy (below). As also mentioned below, there was no precipitation of hTRPA1s and no light scattering (that would indicate the denaturation), and the spectra of the same samples when recorded after couple of hours of incubation at room temperature restored back to the original value.

Taken together, these data clearly show that the effects we monitor are simply due to transient temperature-induced conformational changes in hTRPA1s and not related to basic experimental conditions causing non-specific protein denaturation, pH, stickiness etc.

ii) the purification level of the two variants seems to be quite different (compare the SDS-PAGE in this MS with ref. 7). This further adds to problem i)

Authors: In our previous studies on hTRPA1 and Δ 1-688 hTRPA1 we used *Pichia pastoris* allowing us to produce large amounts of proteins. We are now changing expression system to insect cells (Hi5 and sf9) to optimize and facilitate cryo-EM studies of hTRPA1 with focus on Δ 1-854 hTRPA1. The purity of Δ 1-688 hTRPA1 used here and in previous functional and mass spectrometry studies (Moparthy et al *Sci Rep* 2016; Moparthy et al *Cell Calcium* 2020, Moparthy & Zygmunt *Cell Calcium* 2020; *Int J Med Sci* 2016) was initially determined as 95 % (Moparthy et al *PNAS* 2014) and should not be of any concern. Regarding Δ 1-854 hTRPA1, we reasoned that one step purification without any single substantial impurity would still be acceptable if the channel activity could be blocked by the **selective mammalian TRPA1 antagonist HC030031** (McNamara et al *PNAS* 2007; Gupta et al *Sci Rep* 2016), which is indeed the case. As shown in the gel (now Fig. 1), Δ 1-854 hTRPA1 appears as a strong band at 25 kDa (monomer) and at 50 kDa (dimer). In addition, the band between 50 - 75 kDa most likely indicates post-translational modification of Δ 1-854 hTRPA1. We estimate the purity at least to 80 %. It seems unlikely that the remaining background smear contains enough of any potential Hi5-cell expressing heat-sensitive ion channel mediating the responses. Also, from an evolutionary TRPA1 perspective (for review see Sinica & Vlachova *Physiol Res* 2021), we would have expected no lost heat sensitivity and no **HC030031** block of the Δ 1-854 hTRPA1 currents at 30°C if any endogenous Hi5-TRPA1 mediated these currents in the present study. We are thus confident that trp fluorescence measurements with Δ 1-854 hTRPA1 samples are attributed to the 2 tryptophans present in this protein.

iii) The drop observed upon heat exposure seems to be extremely strong in case of D1-688. Did the authors check for reversibility, loss of protein due to sticking to Eppendorf-tubes etc?

Authors: We apologize for not clarifying this but all incubations were done in the spectrofluorometer, inside the cuvette, with constant steering. No precipitation was observed, no light scattering (that would indicate the denaturation) and the spectra of the same samples when recorded after couple of hours of incubation at room temperature restored back to the original value.

=> Thus, the information content of this part is possibly rather limited, and one might wonder, whether this warrants the extensive discussion addressing this point. The experiment, which most clearly shows a conformational transition is the cold-response of Δ 1-854, showing a sudden drop at around 20°C. Also the cold-response of Δ 1-688 probably reflects some structural changes, since intrinsic thermal response of trp would actually lead to an increase in intensity. However, what has to be also kept in mind, that temperature changes shift the pH, and different constructs might response differentially to this effect.

Authors: We apologize for not having mentioned that these experiments were performed with proteins in PBS, the pH of which is supposed to be stable within this temperature interval. Indeed, we found the pH to be constant over the temp interval used for trp fluorescence studies. This is now included in the methods section. When checking the buffer pH from 4 - 40°C the change varied from 7.30 - 7.34, which we consider to be too low to affect the observed changes.

c) TCEP inhibits cold responses, implying the formation of cys-bridges could be important. Thus, the role of three cysteines were studied in whole-cell patch clamp at 15 and 25°C. Each mutation leads to some effect, but each is different.

=> The main conclusion from this is, that these observations support an allosteric model.

Authors: Please note that the data with whole-cell recordings are replaced with recordings under steady-state conditions as requested by reviewer #1 (point 6). The overall conclusion is still the same.

Thus overall, the findings are interesting, but possibly in case of the trp-fluorescence measurement not as robust as the authors assume. The problems described above should be addressed and mentioned. The discussion should be shortened and straightened. In the present style, it is difficult to follow.

Authors: Because the interpretation of these experiments may not be as straight forward when comparing $\Delta 1-688$ hTRPA1 and $\Delta 1-854$ hTRPA1, it is difficult to avoid the somewhat speculative discussion. Nevertheless, at the end of this paragraph in the discussion, we have added a few lines for the audience to consider when reading this part of the study. "It should be noted that the comparison of $\Delta 1-688$ hTRPA1 and $\Delta 1-854$ hTRPA1 fluorescence may be complicated, since the two remaining tryptophans in $\Delta 1-854$ hTRPA1 that are located within the CTD may experience a different local environment in the absence of VSLD. Nevertheless, our results clearly show that cold and heat responses within the same construct involve different channel conformational changes. Also, there is a substantial rearrangement of the CTD at the interface between warm and cold temperatures as shown in $\Delta 1-854$ hTRPA1."

Specific comments

1) Tryptophan fluorescence studies:

a) The structural changes are followed based of tryptophan fluorescence of detergent solubilized protein. Often the structure of membrane proteins differ in detergent micelles compared to lipid membranes. Are there any studies showing that this approach is valid, and can be extrapolated to the lipid environment?

Authors: To our knowledge, there are only two previous studies linking trp fluorescence and TRP channel bilayer recordings (Billen et al *JBC* 2015; Moparathi et al *Sci Rep* 2016). Considering that the study on TRPV3 was conducted by top-notch TRP channel researchers, we feel confident that our trp fluorescence studies here and previously (Moparathi et al *Sci Rep* 2016) add valuable information on protein conformational changes that cannot be obtained by electrophysiological studies. In addition, it is also encouraging that the same bilayer planar patch-clamp technology, as in our study, was used to study intrinsic TRPV3 activity (Billen et al *JBC* 2015).

b) Furthermore, due to the rather strong temperature dependence of pH in case of Hepes ($dpH/dT = -0.022$) also could add to some of the effects observed. If the pH was set to pH 7.8 at 25°C, the pH will be around 8.2 at 7°C and below 7.5 at 40°C. This should at least be mentioned.

Authors: Please see the answer above.

c) Why is there an emission band around 420 nm in the spectra of heated $\Delta 1-688$? In the cooled spectra this band is missing. Actually, also the shape at the low wavelength side and the overall intensity is different. Are these two different preparations with different levels of purity?

Authors: The emission band around 420 nm could be fluorescence by trp and/or tyr since depending on their environment dramatic emission shifts to higher wavelengths can occur (Vivian & Callis *Biophys J* 2001; Lange et al *Angew Chem Int* 2019; Gorokhov et al *Doklady Biochemistry and Biophysics* 2021). As this is not visible in the full length hTRPA1 and Δ 1-854 hTRPA1 fluorescence spectra (Moparhi et al *Sci Rep* 2016), it is likely contributed by trp/tyr in the VSLD as revealed in the absence of N-ARD, further supporting the important role of VSLD in heat sensation. Δ 1-688 hTRPA1 with a purity of 95 % (Moparhi et al *PNAS* 2014) has also been used in other studies by us using the same bilayer recording assay and mass spectrometry (Moparhi et al *Sci Rep* 2016; Moparhi et al *Cell Calcium* 2020, Moparhi & Zygmunt *Cell Calcium* 2020; *Int J Med Sci* 2016). The amount of protein and the purity of Δ 1-688 hTRPA1 should not be significantly different, although different batches of protein have been used also in the present study.

d) The hump at 305 nm could be water-Raman scattering peak or tyrosine. In case of the former, it should be visible also in other spectra with similar intensities. Are all spectra shown in fig.3 measured at exactly the same settings of the fluorimeter, so that the intensities can be compared? If so, then the hump is probably stemming from tyrosines.

Authors: As mentioned above the protein samples were incubated in the cuvette, inside the spectrofluorometer, as it was equipped with thermal block that allows precise temperature settings and with an electrode immersed in the solution to report the actual temperature of the solution. Once the solution reaches the desired temperature, the spectra get recorded and the new (lower or higher) temperature is set. Therefore, the effect most likely originates from tyrosines. It is also present for the full length hTRPA1 (Moparhi et al *Sci Rep* 2016), now added to the discussion.

2) Patch-Clamp:

The Δ 1-688 mutation leads to higher current density (in magnitude) at negative holding potential, and lower at positive voltage. What about Δ 1-854?

Authors: This has not been tested for Δ 1-854. Here, we restricted our efforts using a hp of +60 mV as this would correspond to -60 mV in the whole-cell configuration mode. However, as discussed below (point 12), it would be of interest to follow up with future studies and more detailed analysis to better understand channel behavior with regard to conductance and pore properties, especially as we have previously shown that the conductance for ligands and pressure is dependent on the N-ARD as well as voltage (Moparhi et al *PNAS* 2014; Moparhi & Zygmunt *Cell Calcium* 2020).

Minor comments

1) When comparing the results of Δ 1-688 and Δ 1-854, the differences are a bit difficult to grab as described in the text. It might help to start from the minimum of activity in each case (22°C and 20°C, respectively) and describe the change in activity by heating or cooling. This is done by reporting Q10 values, but it is not always clear, what the reference temperature is (e.g. legend in Fig. 1, last sentence: Q10 cold 0 56, Heat (20 to 30°C) = 29. Is q10 cold refererering to 10 to 20°C?

Authors: This part of the Results section is now re-written and calculations of Q10 values are clarified including references to previous studies.

2) Why is it stated (results, first paragraph) that Δ 1-854 has lost its heat sensitivity? There is an increase in channel activity between 20 and 25°C. Thus, one observes a strongly decreased heat sensitivity.

Authors: We agree with the reviewer and it is now stated that lost heat sensitivity is within 25 - 40°C.

3) Figure 4: please add, that trps are in blue, and cysteine in yellow. The letters of the amino acids are pretty small and hard to decipher.

Authors: Done.

4) Figure 3: It would be easier for the reader to follow, if the sub-figures were numbered A,B,C etc.

Authors: Done.

5) Why is the temperature range color coded for one mutant, but not for the other? (blue/red areas in fig. 2 and 3)

Authors: Only the transition/intermediate state is now colored to point out this prominent feature not seen in $\Delta 1-688$ or hTRPA1 (Moparathi et al *Sci Rep* 2016).

6) Page 5, second paragraph: "Repeated voltage ramps....in control extracellular solution.." Meaning of control solution unclear.

Authors: New figure and accompanying text (now Fig. 7).

7) Page 5, second paragraph, last third: "C1021A and C1025A were reduced by 0.5 +/- 0.3 ..." Typo in the error?

Authors: New figure accompanying text (now Fig. 7).

8) Page 5, first sentence in discussion: "been shown to response directly"...please clarify "directly". Is this meant in contrast to others, where the thermo-sensing is performed by other proteins, which interact with TRP channels?

Authors: This is now changed to "...but only few thermoTRPs have been shown to intrinsically respond to a change in temperature..."

9) page 7, first paragraph, last third: "nadir" : a typo, I guess.

Authors: Nadir ("the lowest or most unsuccessful point in a situation") is used to define the temperature at which the lowest channel activity appeared.

10) Fig. 7: CTD is referred to as transition state. A transition state is clearly defined (see enzymatics); an alternative would be "intermediate state", maybe.

Authors: This is now changed as suggested.

11) Page 6: second paragraph, middle, discussion of the effect of removing the VSLD, loss of heat sensitivity: "this latter indicating heat sensitive structures within the VSLD or its uncoupling from a heat-sensitive CTD". The first part is clear; but the second part not really. I assume, that the authors want to say, that with intact VSLD, it is coupled to the heat-sensitive CTD, supporting the response to heat? Please clarify.

Authors: It now reads “The latter may indicate that heat sensitive structures are within VSLD or that the heat sensitivity is located to CTD and that the VSLD is supporting CTD in heat-evoked channel gating.”

12) Of note, the single channel conductance of the D1-854 mutant is much less temperature dependent, while the D1-688 has a strong, non-monotonous dependence. This is not mentioned at all.

Authors: Interestingly, pore dilation occurs in TRPA1, indicating that the TRPA1 pore structure is dynamic (Banke et al *Mol Pain* 2009). Maybe the gating of TRPA1 as reflected by less variability in P_o is more robust than its pore properties as reflected by greater variability in conductance. Indeed, the conductance, but not P_o , decreased when hTRPA1 was repeatedly exposed to 15°C (Moparthy et al *PNAS* 2014). It is noteworthy that TRPA1 single-channel conductance values differ greatly depending on stimuli, species, cell and test conditions including ion composition and co-factors such as calcium and polyphosphates (Zygmunt & Högestätt, *Handb Exp Pharmacol* 2014, chapter 5 including Table 1). Although, our hTRPA1s when purified and reconstituted into bilayers may display various single-channel channel opening levels in response to the same temperature as well as the same ligand (Moparthy et al *PNAS* 2014), the conductance values are within those reported for whole-cell configurations as well as isolated cell membrane patches (Zygmunt & Högestätt, *Handb Exp Pharmacol* 2014.) For example, the single-channel conductance value was 40 pS at 15°C compared to 91 pS at 25°C and a hp of +50 mV (Karashima et al *PNAS* 2009). Thus, there is no simple biophysical fingerprint of hTRPA1 with regard to single-channel conductance. In the present study, we have only analyzed the most frequent single-channel conductance state, allowing us to make comparisons with our previous studies. Importantly, this “main” single-channel conductance of Δ 1-854 hTRPA1 is overall similar to “main” hTRPA1 and Δ 1-688 hTRPA1 conductance values within the temperature interval of 25 - 40°C (hp +60 mV), providing evidence that Δ 1-854 truncation did not affect the pore function but the sensitivity to heat. However, the reviewer may be right that VSLD not only affect channel activity as measured by open probability but also pore function as indicated by non-monotonous conductance. Clearly, it is needed to follow up with future studies and more detailed analysis to better understand channel behavior with regard to conductance and pore properties, especially as we have previously shown that the conductance for ligands and pressure is dependent on the N-ARD as well as voltage (Moparthy et al *PNAS* 2014; Moparthy & Zygmunt *Cell Calcium* 2020). Thus, it would not be surprising if the N-ARD intramolecularly also modulates the hTRPA1 pore function in response to temperature, which these results could indicate.

Reviewer #3 (Remarks to the Author):

Authors: We are thankful to the reviewer for the very valuable and constructive criticism on our study. We are happy to know that the reviewer finds the study of interest, and hope that we have satisfactorily addressed below all points raised by the reviewer.

In this manuscript, Moparthy et al. studied the role of different structural domains in the temperature sensitivity of TRPA1. The authors claim that the heat sensitivity of TRPA1 is dependent on the VSLD and CTD and the cold sensitivity is dependent on the CTD and pore domain. In general, these findings are quite interesting and new. However, from my point of view, several flaws hinder this manuscript from being suitable for publication. Below are a few of the most critical issues:

1) The authors only use truncated TRPA1 protein for their lipid bilayer studies. Although the difficulty in purifying the full TRPA1 protein is understood, the authors' conclusion must be supported by demonstrating that in their hands, using the same recording methodologies, the full length and the truncated share the relevant features. Moreover, considering that previous studies showed that

TRPA1 temperature sensitivity requires the N-ARD (the truncated part) (PMID: 21930928), the recording of the full length in lipid bilayer becomes critical for this manuscript.

Authors: We agree that it is key to make comparisons with full length hTRPA1, which has already been done using identical experimental conditions (Moparthy et al *PNAS* 2014, *Sci Rep* 2016) as was also mentioned in the original submitted manuscript version. We have never excluded the N-ARD as important in modulating TRPA1 temperature responses. However, we and others (e.g., Clapham & Miller *PNAS* 2011; Jabba et al *Neuron* 2014; Castillo et al., *Physical Biology* 2018) have noted that mutagenesis and/or chimeras also within the N-ARD could indirectly affect thermoTRP behavior in response to temperatures recognized elsewhere in the proteins. A phenomenon known as functional antagonism. Along with this is our hypothesis that a change in channel redox state can cause conformational changes and thereby hide or reveal temperature sensitive structures (Moparthy et al *Sci Rep* 2016). No doubt, cryo-EM studies together with functional studies such as in the present study will hopefully be the way forward to understand exactly how thermoTRPs are gated.

2) The authors show that the truncated TRPA1 shifted its conductance between 15 and 40 degrees (both at +60mV) by almost a factor of 2. How comes? Do the authors suggest that the pore is only partially open in the cold? These results require explanation and normalization if the authors suggest that its ion mobility issues.

Authors: Interestingly, pore dilation occurs in TRPA1, indicating that the TRPA1 pore structure is dynamic (Banke et al *Mol Pain* 2009). Maybe the gating of TRPA1 as reflected by less variability in P_o is more robust than its pore properties as reflected by greater variability in conductance. Indeed, the conductance, but not P_o , decreased when hTRPA1 was repeatedly exposed to 15°C (Moparthy et al *PNAS* 2014). It is noteworthy that TRPA1 single-channel conductance values differ greatly depending on stimuli, species, cell and test conditions including ion composition and co-factors such as calcium and polyphosphates (Zygmunt & Högestätt, *Handb Exp Pharmacol* 2014, chapter 5 including Table 1). Although, our hTRPA1s when purified and reconstituted into bilayers may display various single-channel channel opening levels in response to the same temperature as well as the same ligand (Moparthy et al *PNAS* 2014), the conductance values are within those reported for whole-cell configurations as well as isolated cell membrane patches (Zygmunt & Högestätt, *Handb Exp Pharmacol* 2014.) For example, the single-channel conductance value was 40 pS at 15°C compared to 91 pS at 25°C and a hp of +50 mV (Karashima et al *PNAS* 2009). Thus, there is no simple biophysical fingerprint of hTRPA1 with regard to single-channel conductance. In the present study, we have only analyzed the most frequent single-channel conductance state, allowing us to make comparisons with our previous studies. Importantly, this “main” single-channel conductance of $\Delta 1$ -854 hTRPA1 is overall similar to “main” hTRPA1 and $\Delta 1$ -688 hTRPA1 conductance values within the temperature interval of 25 - 40°C (hp +60 mV), providing evidence that $\Delta 1$ -854 truncation did not affect the pore function but the sensitivity to heat. Clearly, it is needed to follow up with future studies and more detailed analysis to better understand channel behavior with regard to conductance and pore properties, especially as we have previously shown that the conductance for ligands and pressure is dependent on the N-ARD as well as voltage (Moparthy et al *PNAS* 2014; Moparthy & Zygmunt *Cell Calcium* 2020). Thus, it would not be surprising if the N-ARD intramolecularly also modulates the hTRPA1 pore function in response to temperature, which these results could indicate.

3) It is not clear why several points in the tables are from previous studies. The authors should repeat the experiments or just cite the previous study. The current presentation is misleading. The reader might think that all the data was acquired in the same experimental session.

Authors: It is clearly stated that values are from previous studies. These values were provided as a courtesy to the audience thereby facilitating the comparison between all tested hTRPA1s (Moparthy

et al *PNAS* 2014; Moparthy et al., *Sci Rep* 2016). If the reviewer and editor find it misleading then we will of course remove these values from the Tables.

4) The bars in figure 6E are misleading and do not reflect the presented data in the figure. It is very tough to review this figure due to this problem.

Authors: This figure is now replaced with data based on steady-state conditions as requested by reviewer #1.

5) The single-channel recordings in figure 2C look like multiple channels. Also, the authors should explain and show the Q10 calculation. It seems that Q10 of 1000 should be explained in detail.

Authors: The channel activity is single channel openings of the same magnitude as in Fig. 2a (now Fig. 3a) at 30°C, but with high frequency that perhaps may give the impression of multiple channels. If more channels were recruited then the likelihood of several channels opening at the same time would increase and as a result an increase in channel current levels would be observed, which is not the case in Fig. 2C (now Fig. 3c) and elsewhere. Occasionally, we obtained bilayers with multiple channels. However, they were not used in our studies, as we aimed to keep the level of hTRPA1 “expression” to a minimum to facilitate studies of single-channel activity otherwise complicated by too many channels and macroscopic currents. Here we defined the most frequent single-channel conductance state, and then analyzed channel opening and complete closure regardless of subconductance states. Single-channel open probability was calculated from the time constants, which were derived from the exponential standard of conventionally binned dwell-times. An initial two terms analysis was performed to determine how well the fitting of the model was, after which multiple terms were performed to improve the model fit. All experiments were analyzed using the same conditions to choose the most common channel open probability. Except for analysis of Δ 1-854 hTRPA1 open and closed time, single-channel kinetics were not analyzed in more detail. Our analyses allowed us to make comparisons with our previous studies, in which we have used the same methodological approach and data analysis including Clampfit software 9 (Molecular Devices) to determine single-channel conductance and open probability (Moparthy et al *PNAS* 2014; Moparthy et al *Sci Rep* 2016; Babes et al *J Neurosci* 2016; Survery et al *JBC* 2016; Moparthy et al *Cell Calcium* 2020; Moparthy & Zygmunt *Cell Calcium* 2020). Furthermore, the same approach was used to analyze the purified MscL single-channel mechanosensitivity (Barthmes et al *Eur Biophys J* 2014) as well as the activity of purified hTRPV3 (Billen et al *JBC* 2015), NaV proteins (Shaya et al *PNAS* 2011) and connexin 43 (Cx43) hemichannels (Carnarius et al *JBC* 2012) at a single-channel level. Although, it could be interesting to analyze single-channel open probability for separate conductance levels in response to temperature here and in previous studies, we feel confident that our single-channel activity data are adequately presented to address the aim of the study.

The Q10 values were calculated by the formula given in materials and methods p. 13. Present and previous calculated Q10 values are now explained in detail.

REVIEWER COMMENTS

Reviewer #1 (Remarks to the Author):

This is an interesting study and the authors have significantly improved the manuscript by the inclusion of additional data in Figure 1 and Figure 7.

That said, I remain puzzled by most of the bilayer recordings - repeating my earlier comments (previous points 3-5). The traces often do not seem to present a steady-state condition, and there does not seem to be a comprehensible explanation for the apparently haphazard temperature-dependent variations in single channel conductance or flickering gating behavior, or the odd time course of the RR effect shown in Figure 2c. I would appreciate data that increase the reader's confidence that recordings with single channel conductance of 15 ± 3 pS at 10 degrees, 68 ± 17 pS at 20 degrees, 27 ± 4 pS at 35 degrees and 54 ± 12 pS at 40 degrees actually represent recordings of the same channel.

Reviewer #2 (Remarks to the Author):

In the rebuttal, the authors addressed most of the points raised. Thus the MS can be published, with some minor modifications:

With respect to the trp-experiments, I would clarify the following points

- 1) The extreme decrease in intensity is very untypical; usually upon denaturation the fluorescence intensity increases (after correction for the intrinsic temperature dependence).
 - i) It should be clarified that the hump (= tyrosin) is only visible because the tryptophan fluorescence was quenched that much.
 - ii) I would add the information that the fluorescence changes are reversible; this adds much credibility to the extreme fluorescence changes.
- 2) With respect to the emission at 420 nm:
This value is really found very rarely; I would actually give a reference for this type of emission. Juneja, Shreya, et al. "Unprecedented Intramolecular Association-Induced Fluorescence in Tryptophan-Conjugated Peptidomimetics." *The Journal of Physical Chemistry B* 123.14 (2019): 3112-3117. In this work it is stated, that a trp-trp dimer is responsible for the emission.

Comments with respect to the new version:

Line 113 and 119: wouldn't it be sufficient refer to the equation for Q_{10} with Eq.1, and just define T_1 and T_2 ?

Line 120: "still responded with similar...". Responded to what? Maybe better: "still showed an activity..."

Suppl. Tab.1 one: one line missing (between 15 and 20°). Maybe add how P_o is calculated from the dwell times.

Line 125: better maybe: "at 15°C, the time constant for the open state is 52 to 131 fold higher than at the other temperatures"

Line 130: in this analysis it is assumed that the channel opens stochastically in general, isn't it? Stochastic behavior does not imply a certain transition probability. Therefore, the phrasing is misleading.

Line 256: another possibility for the differences between whole-cell recordings and reconstitute protein is, that upon reconstitution the channel does not adopt its native state completely. Minor and basically undetectable structural changes might well have functional consequences.

Line 290: "some conformational change.. must still occur". Conformational changes relating to channel activity per se? due to the lack of heat-response, it cannot be temperature induced conformational

changes..

Line 308: nadir should be defined somewhere, e.g. in the abbreviations.

Line 309: is there any indication that partial denaturation of the CTD occurs, e.g. circular dichroism ? or is it just speculation? The changes in Trp fluorescence at least do not indicate denaturation processes, since the wavelength of maximum emission does not shift to higher wavelengths.

Reviewer #3 (Remarks to the Author):

Although the authors choose to provide explanations rather than experiments to all the points raised by me (and the other reviewers), I will only focus on one issue that all reviewers raised, and the authors made no effort to solve.

The authors suggest (with words and without providing data) that the single-channel conductance changes they describe are due to pore dilation. They based it on three studies from 2009 to 2011, claiming that TRPA1 also has pore dilation (similar to TRPV1) under continuous activation. Of note, TRPA1 has been rigorously studied from all aspects since its cloning, and the fact that only three studies are dealing with its pore dilation is quite puzzling (doi: 10.1186/1744-8069-5-3; doi: 10.1152/ajpcell.00489.2009; doi: 10.1016/j.brainres.2011.01.021). For comparison, TRPV1 pore dilation is religiously studied from physiology to structure. Nevertheless, the authors' suggestion is still valid, and pore dilation may explain the changes in the conductance. But it must be proven by experiments. The authors should check the permeability changes shown in the paper they cite in the rebuttal and demonstrate that pore dilation is the reason. Otherwise, this manuscript is not suitable for publication. This manuscript's conclusions are based on single-channel data showing a dramatic change in conductance with no supporting experiment, resulting in a preliminary study.

REVIEWER COMMENTS

Reviewer #1 (Remarks to the Author):

This is an interesting study and the authors have significantly improved the manuscript by the inclusion of additional data in Figure 1 and Figure 7.

Authors: We appreciate very much the time and effort that this reviewer has put into the revision of our study. The reviewer's thorough and constructive criticism is most valuable. We are happy to know that the reviewer finds the study of interest, and hope that we now have satisfactorily addressed below all points raised by the reviewer.

That said, I remain puzzled by most of the bilayer recordings - repeating my earlier comments (previous points 3-5). The traces often do not seem to present a steady-state condition, and there does not seem to be a comprehensible explanation for the apparently haphazard temperature-dependent variations in single channel conductance or flickering gating behavior, or the odd time course of the RR effect shown in Figure 2c. I would appreciate data that increase the reader's confidence that recordings with single channel conductance of 15 ± 3 pS at 10 degrees, 68 ± 17 pS at 20 degrees, 27 ± 4 pS at 35 degrees and 54 ± 12 pS at 40 degrees actually represent recordings of the same channel.

We apologize for not having addressed satisfactorily the above points raised by the reviewer. The traces in Figs. 2 and 3 are part of longer recordings used for reliably determining single-channel open probability and chosen to fairly illustrate in a single short trace that channel activity is not monotonous over the entire recording time. Thus, different patterns of activity occurred within the entire period of analysis exceeding the 10 s traces shown in Figs. 2 and 3 (see for example the $\Delta 1$ -854 hTRPA1 trace at 40 °C in Fig. 3a, which is taken from a longer recording and marked by the red dotted box in the trace below). Some of the traces therefore include an intermittent period of low or no activity. Furthermore, even at the same temperature the channel may respond with both long-lasting openings/closures as well as flickering behavior. This intriguing channel behavior is further substantiated by additional data/recordings shown in Supplementary Figs. 2-6. At this stage, we can only compare our recordings with those performed in studies demonstrating similar channel behavior for purified TRPV1, TRPM8 and TRPM3 but unfortunately these authors also did not provide an explanation for such mixed channel gating behavior (Zakharian et al JNS 2010; Sun & Zakharian JBC 2015; Uchida et al FASEB J 2016). To better understand this, future in-depth biophysical studies, similar to those performed with heterologously TRPV1 (Hui et al Biophys J 2003; Liu et al Biophys J 2003), are needed. However, such extensive studies on our purified hTRPA1s, for each of the chosen temperatures within 10-40 °C, would need sampling of much more data to reach any reliable conclusion about the gating mechanism involved in this mixed channel behavior. That could, based on our experience, take years to complete considering that bilayer patch-clamp single-channels studies involve unique critical and time-consuming steps including expression/purification and channel incorporation into the lipid bilayer of the various TRPA1s. In fact, one key limitation is the channel incorporation into the lipid bilayer with an estimated overall success rate of only 30%.

Regarding the single-channel conductance, we have performed a more detailed characterization of this channel property at the chosen temperatures in the present study (Supplementary Figs. 5 and 6). Importantly, we observed a similar main conductance state as well as several overlapping sub-conductance states for $\Delta 1-688$ and $\Delta 1-854$ hTRPA1 at both cold and warm temperatures. The non-monotonous conductance values (Table 1 and 2) were due to grouping individual experiments without considering the variation in single-channel conductance state. Thus, we have removed the conductance values in the previous Table 1 and 2, both of which are replaced by the new Table 1 only containing Po values, the analysis of which appropriately includes the main and any sub-conductance state, for $\Delta 1-688$ and $\Delta 1-854$ hTRPA1. Nevertheless, it may be interesting that both purified hTRPA1 and heterologously expressed mouse TRPA1 conductance is lower at 15 °C and below (Moparthi et al PNAS 2014, Sci Rep 2016; Sawada et al Brain Res 2007, Karashima et al PNAS 2009), and in the present study channel openings at 15 °C seem to consist of mostly low sub-conductance states for both $\Delta 1-688$ hTRPA1 and $\Delta 1-854$ hTRPA1. However, further separate studies are needed to fully characterize the various single-channel conductance states including the frequency of occurrence with changes in temperature.

A paragraph on the above raised issues is now included in the results section. In this paragraph, we also mention that we and others, when performing bilayer studies of purified TRP channels (Zakharian et al JNS 2010; Sun & Zakharian JBC 2015; Uchida et al FASEB J 2016), share the challenge to capture channel activity at all temperatures in the same membrane patch (Supplementary Figs. 2 and 3).

Regarding the time course of the RR effect, we can only conclude that this is similar to our previous studies (Moparthi et al Sci Rep 2016; Babes et al JNS 2016). In previous organ bath studies, we have noticed that RR stained the plastic material and it could be that unspecific RR binding has to be saturated before the channel inhibitory effect appears also in the present study? However, as we provide solid data with the selective TRPA1 inhibitor HC030031, we have decided to remove the RR data from Fig. 2 to avoid confusion and the need to justify a possible cause for the delayed onset of blockade.

Taken together, we hope that the reviewer agrees that we now report reliably a unique relationship between structure and temperature-dependent activation of hTRPA1 by the appropriate analysis of **single-channel open probability** and tryptophan fluorescence conformational changes of purified TRPA1. We also link the temperature sensitivity to the redox state of hTRPA1 and our whole-cell recordings allow us to pinpoint cysteines outside the N-ARD and in close vicinity of the lower channel gate as critical for both the temperature- and voltage-dependent gating of hTRPA1.

Reviewer #2 (Remarks to the Author):

In the rebuttal, the authors addressed most of the points raised.
Thus the MS can be published, with some minor modifications:

Authors: We appreciate very much the time and effort that this reviewer has put into the revision of our study. The reviewer's thorough and constructive criticism is most valuable. We are happy to know that the reviewer finds the study publishable after having addressed below all points raised by the reviewer.

With respect to the trp-experiments, I would clarify the following points

1) The extreme decrease in intensity is very untypical; usually upon denaturation the fluorescence intensity increases (after correction for the intrinsic temperature dependence).

i) It should be clarified that the hump (= tyrosin) is only visible because the tryptophan fluorescence was quenched that much.

ii) I would add the information that the fluorescence changes are reversible; this adds much credibility to the extreme fluorescence changes.

2) With respect to the emission at 420 nm:

This value is really found very rarely; I would actually give a reference for this type of emission.

Juneja, Shreya, et al. "Unprecedented Intramolecular Association-Induced Fluorescence in Tryptophan-Conjugated Peptidomimetics." *The Journal of Physical Chemistry B* 123.14 (2019): 3112-3117. In this work it is stated, that a trp-trp dimer is responsible for the emission.

Authors: Thank you! It now reads as follows: Notably, the “hump” at 305 nm in the $\Delta 1$ -688 hTRPA1 heat fluorescence spectrum (Fig. 4b) is most likely fluorescence originating from tyrosines, and only visible because of the substantial quenching of the tryptophan fluorescence signal. The small but significant emission at 420 nm for $\Delta 1$ -688 hTRPA1 (Fig. 4b) and possibly $\Delta 1$ -854 hTRPA1 (Fig. 4c), could be related to fluorescence by tryptophans and/or tyrosines, since depending on their environment dramatic emission shifts to higher wavelengths can occur³⁸⁻⁴⁰. The spectra of the samples restored back to the original value when recorded after a few hours at room temperature.”

Comments with respect to the new version:

Line 113 and 119: wouldn't it be sufficient refer to the equation for Q10 with Eq.1, and just define T1 and T2?

Authors: Agree and changed accordingly.

Line 120: “still responded with similar...”. Responded to what? Maybe better: “still showed an activity...”

Authors: Agree. It now reads as follows: “Although, the heat sensitivity within 25 - 40 °C was lost, $\Delta 1$ -854 hTRPA1 still displayed activity with similar single-channel open probability within this temperature interval (Fig. 3a).”

Suppl. Tab.1 one: one line missing (between 15 and 20°). Maybe add how Po is calculated from the dwell times.

Authors: Done.

Line 125: better maybe: “at 15°C, the time constant for the open state is 52 to 131 fold higher than at the other temperatures”

Authors: Agree. It now reads as follows: “At 15 °C, the time constant for the open state is 52 to 131 fold higher than at the other temperatures (Supplementary Table 1).”

Line 130: in this analysis it is assumed that the channel opens stochastically in general, isn't it? Stochastic behavior does not imply a certain transition probability. Therefore, the phrasing is misleading.

Authors: Agree. It now reads as follows: “The rather similar time constant for closed state within 25 - 40 °C as well as the open state within 25 - 40 °C provide evidence that the channel pore opening within this temperature interval is temperature-independent.”

Line 256: another possibility for the differences between whole-cell recordings and reconstitute protein is, that upon reconstitution the channel does not adopt its native state completely. Minor and basically undetectable structural changes might well have functional consequences.

Authors: Added. It now reads as follows: "In addition, it is possible that the reconstituted channel does not adopt its native state completely."

Line 290: "some conformational change.. must still occur". Conformational changes relating to channel activity per se? due to the lack of heat-response, it cannot be temperature induced conformational changes..

Authors: Agree. It now reads as follows: "Notably, some conformational changes of the pore region must still occur in the absence of VSLD within 25 - 40 °C, but independent of changes in temperature as the single-channel open probability was similar within this temperature range."

Line 308: nadir should be defined somewhere, e.g. in the abbreviations.

Authors: Nadir is removed. It now reads as follows: "." This fits well with the lowest single-channel open probability obtained at 20 - 22 °C..."

Line 309: is there any indication that partial denaturation of the CTD occurs, e.g. circular dichroism ? or is it just speculation? The changes in Trp fluorescence at least do not indicate denaturation processes, since the wavelength of maximum emission does not shift to higher wavelengths.

Authors: Maybe partial denaturation could happen although not visible in our CTD? However, "denaturation" has been replaced by "... cold- and heat-induced conformational changes of CTD..."

Reviewer #3 (Remarks to the Author):

Although the authors choose to provide explanations rather than experiments to all the points raised by me (and the other reviewers), I will only focus on one issue that all reviewers raised, and the authors made no effort to solve.

The authors suggest (with words and without providing data) that the single-channel conductance changes they describe are due to pore dilation. They based it on three studies from 2009 to 2011, claiming that TRPA1 also has pore dilation (similar to TRPV1) under continuous activation. Of note, TRPA1 has been rigorously studied from all aspects since its cloning, and the fact that only three studies are dealing with its pore dilation is quite puzzling (doi: 10.1186/1744-8069-5-3; doi: 10.1152/ajpcell.00489.2009; doi: 10.1016/j.brainres.2011.01.021). For comparison, TRPV1 pore dilation is religiously studied from physiology to structure. Nevertheless, the authors' suggestion is still valid, and pore dilation may explain the changes in the conductance. But it must be proven by experiments. The authors should check the permeability changes shown in the paper they cite in the rebuttal and demonstrate that pore dilation is the reason. Otherwise, this manuscript is not suitable for publication. This manuscript's conclusions are based on single-channel data showing a dramatic change in conductance with no supporting experiment, resulting in a preliminary study.

Authors: We thank the reviewer for suggesting a closer look into the issue of single-channel conductance. We have performed a more detailed characterization of this channel property (Supplementary Figs. 5 and 6) clearly showing several overlapping single-channel conductance states for $\Delta 1-688$ and $\Delta 1-854$ hTRPA1 of which the main conductance had a similar conductance at cold (15 °C and 17 °C) and warm temperatures (30 °C - 40 °C) for $\Delta 1-688$ hTRPA1 (55 - 65 pS) and $\Delta 1-854$

hTRPA1 (46 - 58 pS). The non-monotonous conductance values (Table 1 and 2) were due to grouping individual experiments without considering the variation in single-channel conductance state. Thus, we have removed the conductance values in the previous Table 1 and 2, both of which are replaced by the new Table 1 only containing P_o values, the analysis of which appropriately includes the main and any sub-conductance state, for $\Delta 1-688$ and $\Delta 1-854$ hTRPA1. Nevertheless, it may be interesting that both purified hTRPA1 and heterologously expressed mouse TRPA1 conductance is lower at 15 °C and below (Moparthi et al PNAS 2014, Sci Rep 2016; Sawada et al Brain Res 2007, Karashima et al PNAS 2009), and in the present study channel openings at 15 °C seem to consist of mostly low sub-conductance states for both $\Delta 1-688$ hTRPA1 and $\Delta 1-854$ hTRPA1. Thus, we share the reviewer's interest in better understanding how temperature affects single-channel conductance. However, this has to be the basis of a separate in-depth biophysical study of the purified hTRPA1s to fully characterize the various single-channel conductance states including the frequency of occurrence and if/how pore dilation is related to changes in temperature. Nevertheless, regardless of the outcome of such studies, we clearly demonstrate a unique relationship between structure and temperature-dependent activation of hTRPA1 by the appropriate analysis of **single-channel open probability** and tryptophan fluorescence conformational changes of purified TRPA1. We also link the temperature sensitivity to the redox state of hTRPA1 and our whole-cell recordings allow us to pinpoint cysteines outside the N-ARD and in close vicinity of the lower channel gate as critical for both the temperature- and voltage-dependent gating of hTRPA1.

REVIEWERS' COMMENTS

Reviewer #1 (Remarks to the Author):

My main criticism on earlier versions of the manuscript (see also reviewer #3) related to the non-monotonous changes in single-channel conductance with temperature. In the present version, the authors attribute these non-monotonous changes to the existence of subconductance states. Evidence for such subconductance states is provided by some example traces in Supplementary Figures 5 and 6, albeit without any analytical treatment (not even a histogram). This remains the weaker part of an otherwise interesting manuscript.

Reviewer #3 (Remarks to the Author):

I agree with the authors that performing a complete biophysical study of their constructs is time- and labor-intensive. Nevertheless, the authors focused only on the single-channel open probability and omitted the conductance issues. The authors' decision is quite puzzling, considering that both parameters are obtained in the same experiment. How do the authors explain that they have enough reliable data to determine changes in P_o (for each temperature) but require much more data for the conductance? Moreover, it is unclear how the conclusions remain the same, although previous data that the authors highly relied on is now less critical (deleting the conductance parameters from Table 1).

As I mentioned in my first review, this manuscript describes important results, and its novelty is apparent. However, the conclusions of this manuscript are much more dependent on the single channel analysis than any other analysis (e.g., the TRP analysis). Thus, the conclusions should be reflected by the results. Relying only on P_o should also be reflected in the manuscript conclusions. If the authors decide to continue in the current form of the manuscript (i.e., not adding new experiments), the authors should tune down the conclusions appropriately. Thus, considering the rebuttal letter, I suggest revising the text to align the data with the conclusions.

REVIEWERS' COMMENTS

Reviewer #1 (Remarks to the Author):

My main criticism on earlier versions of the manuscript (see also reviewer #3) related to the non-monotonous changes in single-channel conductance with temperature. In the present version, the authors attribute these non-monotonous changes to the existence of subconductance states. Evidence for such subconductance states is provided by some example traces in Supplementary Figures 5 and 6, albeit without any analytical treatment (not even a histogram). This remains the weaker part of an otherwise interesting manuscript.

Reviewer #3 (Remarks to the Author):

I agree with the authors that performing a complete biophysical study of their constructs is time- and labor-intensive. Nevertheless, the authors focused only on the single-channel open probability and omitted the conductance issues. The authors' decision is quite puzzling, considering that both parameters are obtained in the same experiment. How do the authors explain that they have enough reliable data to determine changes in P_o (for each temperature) but require much more data for the conductance? Moreover, it is unclear how the conclusions remain the same, although previous data that the authors highly relied on is now less critical (deleting the conductance parameters from Table 1).

As I mentioned in my first review, this manuscript describes important results, and its novelty is apparent. However, the conclusions of this manuscript are much more dependent on the single channel analysis than any other analysis (e.g., the TRP analysis). Thus, the conclusions should be reflected by the results. Relying only on P_o should also be reflected in the manuscript conclusions. If the authors decide to continue in the current form of the manuscript (i.e., not adding new experiments), the authors should tune down the conclusions appropriately. Thus, considering the rebuttal letter, I suggest revising the text to align the data with the conclusions.

Authors reply to reviewer #1 and #3: We agree with reviewers that single-channel conductance is also an important feature of TRPA1 channel gating. Thus, we now show in representative traces (Figs 4 and 5) that several overlapping single-channel conductance states for $\Delta 1-688$ and $\Delta 1-854$ hTRPA1 of which the main conductance had a similar conductance at cold (15 °C and 17 °C) and warm temperatures (30 °C - 40 °C) for $\Delta 1-688$ hTRPA1 (55 - 65 pS) and $\Delta 1-854$ hTRPA1 (46 - 58 pS). We have moved Supplementary Figures 5 and 6 in the previous Supplementary information into the current main manuscript as Figures 4 and 5, and with corresponding histograms. However, as concluded by Sawada and Karashima (Sawada et al Brain Res 2007; Karashima et al PNAS 2009) as well as supported by the Patapoutian group (del Camino JNS 2010), the single-channel open probability (P_o) but not single-channel conductance (G_s) is the appropriate indicator of TRPA1 temperature-sensitivity. These authors demonstrated that a dramatic change in mouse TRPA1 single-channel open probability with little change in single-channel conductance underlies the TRPA1 cold-sensitivity with a Q10 value ~ 0.1 . In our study, the analysis of P_o is not restricted

to any particular conductance level, and therefore the number of independent experiments performed here is enough for a reliable assessment of channel activity in response to changes in temperature. To dissect the many possible conductance levels and their distribution at all the temperatures studied would need much more recordings to statistically establish any difference in temperature-dependent conductance-level appearance. Still, that would not change the conclusion as P_o is an appropriate indicator also of human TRPA1 temperature-sensitivity (line 172 to 176). The key focus on single-channel analysis is also stressed in the Discussion (line 294). However, in addition to P_o analysis, our conclusion is also based on tryptophan fluorescence studies and whole cell recordings. Please see modified conclusion line 533 to 539.